# Local actin nucleation tunes centrosomal microtubule nucleation during passage through mitosis

Francesca Farina[1,2,3,4†], Nitya Ramkumar[1,2,*,†] (iD), Louise Brown[5], Dureen Samandar Eweis[1,2,§],
Jannis Anstatt[1,2], Thomas Waring[5], Jessica Bithell[5], Giorgio Scita[3,6] (iD), Manuel Thery[7] (iD),
Laurent Blanchoin[4] (iD), Tobias Zech[5] & Buzz Baum[1,2,**]

## Abstract

Cells going through mitosis undergo precisely timed changes in cell shape and organisation, which serve to ensure the fair partitioning of cellular components into the two daughter cells. These structural changes are driven by changes in actin filament and microtubule dynamics and organisation. While most evidence suggests that the two cytoskeletal systems are remodelled in parallel during mitosis, recent work in interphase cells has implicated the centrosome in both microtubule and actin nucleation, suggesting the potential for regulatory crosstalk between the two systems. Here, by using both *in vitro* and *in vivo* assays to study centrosomal actin nucleation as cells pass through mitosis, we show that mitotic exit is accompanied by a burst in cytoplasmic actin filament formation that depends on WASH and the Arp2/3 complex. This leads to the accumulation of actin around centrosomes as cells enter anaphase and to a corresponding reduction in the density of centrosomal microtubules. Taken together, these data suggest that the mitotic regulation of centrosomal WASH and the Arp2/3 complex controls local actin nucleation, which may function to tune the levels of centrosomal microtubules during passage through mitosis.

**Keywords** Arp2/3 complex; centrosomal actin; mitosis; WASH complex
**Subject Categories** Cell Adhesion, Polarity & Cytoskeleton; Cell Cycle
**The EMBO Journal (2019) 38: e99843**

See also: **D Inoue *et al*** (June 2019)

## Introduction

The microtubule (Zhai *et al*, 1996; Meraldi & Nigg, 2002) and actin cytoskeletons (Ramkumar & Baum, 2016) undergo profound parallel changes in dynamics and organisation as cells go through mitosis. These changes play a vital role in the control of animal cell division and begin as cells enter prophase. At this time, the interphase microtubule cytoskeleton is disassembled (Centonze & Borisy, 1990; Niethammer *et al*, 2007; Mchedlishvili *et al*, 2018), allowing microtubule nucleation to become focused at centrosomes (Zhai *et al*, 1996; Piehl *et al*, 2004; Mchedlishvili *et al*, 2018), where gamma-tubulin accumulates (Khodjakov & Rieder, 1999; Bettencourt-Dias & Glover, 2007; Sulimenko *et al*, 2017). With the loss of the nuclear/cytoplasmic compartment barrier at the onset of prometaphase, this is followed by a sudden change in microtubule organisation (Mchedlishvili *et al*, 2018) and dynamics (Zhai *et al*, 1996). During prometaphase, the short, dynamic centrosomal microtubules that remain capture chromosomes (Mitchison & Kirschner, 1985) drive bipolar spindle formation (Magidson *et al*, 2011) and interact with the cortex to guide positioning of the mitotic spindle (McNally, 2013).

The actin cytoskeleton also undergoes changes over the same period. These begin in prophase when the interphase actin cytoskeleton is disassembled (Matthews *et al*, 2012). This likely frees up a pool of actin monomers (Kaur *et al*, 2014), which is then used to assemble a thin (Clark *et al*, 2013), mechanically rigid (Fischer-Friedrich *et al*, 2015), cortical actomyosin network that drives mitotic rounding (Reinsch & Karsenti, 1994; Ragkousi & Gibson, 2014; Sorce *et al*, 2015). While the mechanisms underlying this mitotic switch in actin organisation are not well

1 MRC-LMCB, UCL, London, UK
2 IPLS, UCL, London, UK
3 IFOM, the FIRC Institute of Molecular Oncology, University of Milan, Milan, Italy
4 University of Grenoble, Grenoble, France
5 Institute of Translational Medicine, Cellular and Molecular Physiology, University of Liverpool, Liverpool, UK
6 Department of Oncology and Hemato-Oncology, University of Milan, Milan, Italy
7 Hospital Saint-Louis, Paris, France
 *Corresponding author. Tel: +44 20 7679 3040; E-mail: n.ramkumar@ucl.ac.uk
 **Corresponding author. Tel: +44 20 7679 3040; E-mail: b.baum@ucl.ac.uk
 †These authors contributed equally to this work
 §Correction added on 3 June 2019 after first online publication: Author name was corrected from Durren Samander-Eweis to Dureen Samandar Eweis

understood, the process likely involves the following: (i) the loss of interphase focal adhesions (Dix *et al*, 2018; Lock *et al*, 2018), (ii) the loss of Arp2/3-dependent lamellipodia (Ibarra *et al*, 2005; Bovellan *et al*, 2014; Rosa *et al*, 2015) and (iii) the activation of formins downstream of Ect2/Pbl and the GTPase Rho (Maddox & Burridge, 2003; Matthews *et al*, 2012; Rosa *et al*, 2015; Chugh *et al*, 2017).

Interestingly, these parallel changes in actin and microtubule organisation appear to be largely independent of one another (Mchedlishvili *et al*, 2018). Thus, mitotic rounding is not much altered in cells entering mitosis without microtubules. Conversely, mitotic spindle assembly occurs with relatively normal kinetics in spherical cells that have been treated with latrunculin to remove their actomyosin cortex (Lancaster *et al*, 2013). Thus, during mitotic entry, the two systems appear to be independently regulated. However, this changes at anaphase, where the behaviour of the two filament systems is tightly coordinated for proper cell division. The signals emanating from the anaphase spindle polarises the overlying actomyosin cortex (Rappaport, 1996). This is achieved mainly through the activity of the centralspindlin complex (White & Glotzer, 2012), which binds overlapping microtubules at the midzone. This complex, in turn, recruits Ect2 (Yüce *et al*, 2005; Su *et al*, 2011), leading to Rho activation and assembly of a contractile actomyosin ring (Rappaport, 1996; Fededa & Gerlich, 2012), which drives cytokinesis (Wagner & Glotzer, 2016; Basant & Glotzer, 2018). At the same time, as the anaphase spindle elongates, signals associated with the anaphase chromatin appear to aid relaxation of the polar cortical actomyosin network (Salmon & Wolniak, 1990; Motegi *et al*, 2006; von Dassow, 2009; Ramkumar & Baum, 2016). This leads to the de-phosphorylation of ERM proteins, which cross-link actin to the plasma membrane (Rodrigues *et al*, 2015), to the loss of anillin (Kiyomitsu & Cheeseman, 2013) and to activation of SCAR/WAVE and the Arp2/3 complex at opposing cell poles (Zhang & Robinson, 2005; King *et al*, 2010; Nezis *et al*, 2010; Bastos *et al*, 2012; Luo *et al*, 2014). In some instances, the process of polar relaxation and cell re-spreading is sufficient to drive division in cells that lack an actomyosin ring (Neujahr *et al*, 1997; Dix *et al*, 2018).

In studies looking at the role of the actin cytoskeleton in division, the mitotic cortical actomyosin network has been subject to most scrutiny. This is because cortical cytoskeleton controls animal cell shape and is by far the brightest actin-based structure visible under the microscope. However, two groups have reported the existence of dynamic cytoplasmic actin-based structures in dividing HeLa cells (Mitsushima *et al*, 2010; Field & Lénárt, 2011; Fink *et al*, 2011). While the precise function of this pool of cytoplasmic actin remains unclear, it has been reported to play a role in spindle assembly and positioning in various systems (Woolner *et al*, 2008; Sabino *et al*, 2015). In addition, cytoplasmic actin appears to work together with an unconventional Myosin, Myo19, to aid the partitioning of mitochondria at anaphase (Rohn *et al*, 2014).

Here, building on a previous study that identified the WASH/Arp2/3-dependent nucleation of actin at centrosomes in interphase cells (Farina *et al*, 2016), we have re-examined the dynamics and potential function of non-cortical actin at mitotic exit. Using a combination of cell biology and biochemistry, we report the identification of a pool of WASH/Arp2/3-dependent cytoplasmic actin that is nucleated around centrosomes in early anaphase, which appears to limit the nucleation of centrosomal microtubules.

# Results

In order to explore the possibility that actin is nucleated at centrosomes during mitotic exit, as it is in interphase cells (Farina *et al*, 2016), we fixed a population of HeLa cells and examined the amount of F-actin (phalloidin) and microtubules in a region close to centrosomes at different cell cycle stages. This revealed an increase in the density of F-actin in a small region around the centrosomes during the passage from metaphase to early anaphase (Fig 1A and B). During this period, we observed no significant changes in the levels of non-centrosomal cytoplasmic actin (Fig EV1A and C). This increase in centrosomally associated actin was accompanied by a decrease in microtubules intensity in the same region (Figs 1A and EV1B). A similar increase in actin accumulation and a corresponding decrease in tubulin intensity were also observed around the centrosomes of early anaphase Jurkat cells, a T-lymphocyte cell line (Fig EV1E and F). To investigate the dynamics of actin during this period, we used a spinning disc confocal to image HeLa cells expressing Lifeact-GFP. We observed a dynamic pool of cytoplasmic actin in metaphase, as described previously (Mitsushima *et al*, 2010; Fink *et al*, 2011). At anaphase, this pool became concentrated sub-cortically around the opposing poles (Figs 1C and EV2, Movies EV1–EV3). In order to define the dynamics of actin accumulation around the centrosomes, we generated a stable HeLa cell line expressing RFP-Lifeact and GFP-tubulin and performed relatively high temporal resolution acquisition (Fig 1D). Shortly after anaphase onset, we observed an increase in the levels of actin around centrosomes (Fig 1D–F), while non-centrosomal cytoplasmic actin levels remained unchanged (Fig EV1A and D). This burst of actin filament formation was extremely transient, occurring within minutes of anaphase onset, and was over by the time the cytokinetic furrow became clearly visible (Figs 1C–F and EV2, Movies EV1–EV3). Further, we observed this transient accumulation of actin around centrosomes during anaphase using diverse actin reporters (siR-actin, Lifeact-GFP and RFP-Lifeact), albeit with different accumulation dynamics that we attribute to the nature of the probes and their differing actin-binding kinetics (Figs 1 and EV2). During the same period, the density of microtubules, measured as an integrated intensity in a small region around spindle poles, dropped (Fig 1D and G). Thus, this transient appearance of cytoplasmic actin close to centrosomes at mitotic exit is associated with a reduction in the density of centrosomal microtubules.

In order to better visualise centrosomal actin *in vivo*, and as a method by which to isolate centrosomes for the *in vitro* experiments (see below), we also carried out similar analysis during monopolar cytokinesis (Hu *et al*, 2008). For these experiments, we arrested HeLa cells in prometaphase using STLC (Mayer *et al*, 1999; DeBonis *et al*, 2004)—a treatment that inhibits the Eg5 motor to prevent the assembly of a bipolar spindle (Fig 2). Actin nucleation was then followed as these prometaphase-arrested cells were forced to exit mitosis through the addition of a Cdk1 inhibitor (RO-3306) (Hu *et al*, 2008). Importantly, under these conditions, the analysis of centrosomal actin is facilitated by the fact that the monopole contains both centrosomes and remains far from the cell cortex (Fig 2A)—even though many other aspects of cytokinesis appear similar (Karayel *et al*, 2018). In this experiment, we observed little centrosomal-associated actin in the cells arrested in prometaphase with STLC (Figs 2A and EV3, Movie EV4). However, within ~6 min

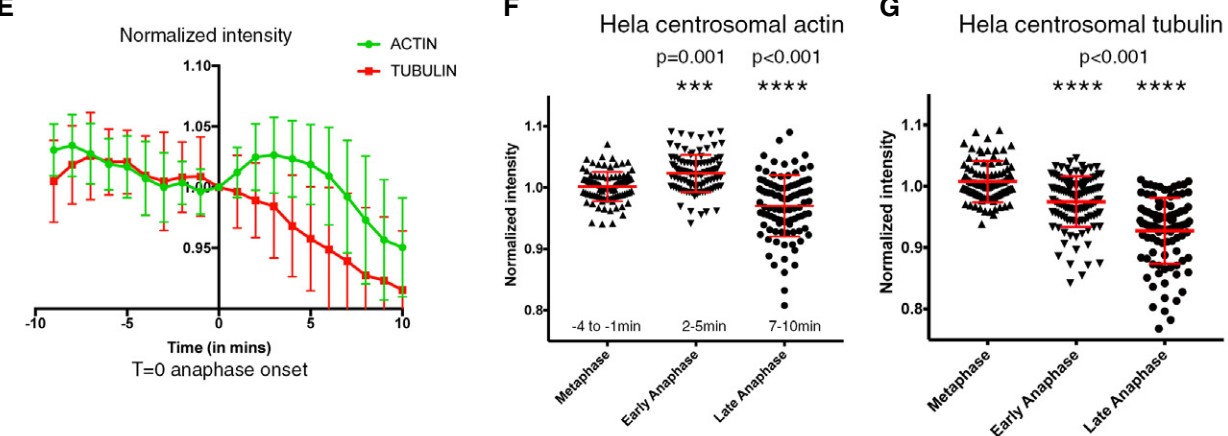

Figure 1.

**Figure 1.  Dynamics of actin and microtubule networks during mitotic exit.**

A   Maximum projection (2 z-slices) of HeLa cells immunostained for F-actin (phalloidin), tubulin, pericentrin and DAPI at metaphase and early anaphase showing actin accumulation around centrosomes in early anaphase. Scale bar = 10 μm

B   Quantification of F-actin (phalloidin) intensity around pericentrin-positive centrosomes in HeLa cells immunostained as in (A), showing the increase in F-actin around centrosomes at early anaphase. Mean actin metaphase = 1 ± 0.03996, n = 92; mean actin anaphase = 1.875 ± 0.08895, n = 121; Student's t-test, ****P < 0.0001.

C   Time-lapse sequence from a representative HeLa cell expressing Lifeact-GFP transiting from metaphase to anaphase, showing accumulation of actin in the presumptive centrosomal region, red arrows. Images represent 4z-projection. Scale bar = 10 μm.

D   Time-lapse sequence of a representative HeLa cell expressing GFP-alpha-tubulin and RFP-Lifeact transiting from metaphase to anaphase showing the changes in actin and microtubules. Images represent 4 z-projection of movies taken every 1 min. Bottom—higher magnification view of the centrosomes C1 and C2 from time lapse above showing a transient increase in actin around the centrosomes in early anaphase. Scale bar = 10 μm, 1 μm in zoom. T = 0 is one frame before anaphase onset. Dotted circle shows centrosome position.

E   Integrated fluorescence intensity of GFP-alpha-tubulin and RFP-Lifeact around the centrosomes for a population of cells exiting mitosis, showing actin accumulation around the centrosomes and a simultaneous decrease in tubulin over time. N = 30 centrosomes (15 cells). T = 0 is one frame before anaphase onset. Graph shows mean ± SD, and values were normalised to intensity at T = 0.

F   Quantification and comparison of actin (RFP-Lifeact) and tubulin (a-tubulin-GFP) fluorescence intensity around the centrosomes at metaphase (−4 to −1 min), early anaphase (2–5 min) and late anaphase (7–10 min) for a population of cells. Mean actin intensity at metaphase = 1.002 ± 0.0239, n = 97; early anaphase = 1.023 ± 0.03047, n = 101; late anaphase = 0.9703 ± 0.05004, n = 98. Tubulin intensity at metaphase = 1.008 ± 0.03356, n = 106; early anaphase = 0.9748 ± 0.04124, n = 116; late anaphase = 0.9276 ± 0.05382, n = 101; one-way ANOVA.

of Cdk1 inhibitor addition we observed a burst of actin filament formation close to centrosomes (Figs 2A and EV3, Movie EV4). Strikingly, these actin filaments formed parallel bundles that lay in between the astral microtubules emanating from the large monopole present in these cells (Fig 2A and B). To confirm this finding, cells arrested in prometaphase and cells after forced exit were fixed and stained to visualise actin filaments (phalloidin). We observed a similar increase in actin around centrosomes during forced exit in multiple cell lines: in HeLa cells (Fig 2C–E), the Jurkat T-cell line (Fig EV4A and B) and the MAVER1 B-cell line (Fig EV4C and D). Further, this actin accumulation was accompanied by a reduction in the density of centrosomal microtubules (Fig 2E). Thus, cytoplasmic actin appears to transiently accumulate around centrosomes during mitotic exit in both monopolar and bipolar divisions, and this is accompanied by a local decrease in microtubule density.

Previous work demonstrated that the actin formed at interphase centrosomes is nucleated by a local pool of Arp2/3 (Farina et al, 2016). To investigate the role of Arp2/3 complex in actin accumulation during mitotic exit, we treated cells with either DMSO or the Arp2/3 complex inhibitor, CK666, and determined the amount of centrosomal actin in fixed (Fig 3) and live cells (Fig EV5). While the DMSO control behaved as described above, the CK666 Arp2/3 inhibitor was effective in eliminating the formation of cytoplasmic actin, including the actin emanating from the centrosome (Fig 3A–D). This was the case in both regular bipolar mitosis (Fig 3A and B), forced monopolar exit (Fig 3C and D) and live monopolar exit (Fig EV5A and B). While the actin accumulation decreased in cells treated with the Arp2/3 inhibitor, interestingly we found that Arp2/3 inhibition

prevented the reduction in the density of microtubules associated with centrosomes during normal mitotic exit (Figs 3E and EV6A) and in monopolar exit (fixed cells–Figs 3F and 6B; live cells—EV5C). This suggests that the pool of actin associated with the centrosomes may influence centrosomal microtubules in early anaphase. This could aid division, as we observed a small percentage of cells with spindle oscillations (data not shown).

These results indicate the role of Arp2/3 in generating the burst of cytoplasmic actin associated with centrosomes at mitotic exit. To determine the localisation dynamics of Arp2/3 during mitotic progression, we fixed and stained cells using an antibody against a component of the complex, p34 (Fig 4A). This revealed a pool of Arp2/3 at the centrosome, marked by centrin 1 (Fig 4A and B) that was very low to undetectable in metaphase, but increased significantly in intensity as cells were forced to leave mitosis (Fig 4A and B). A similar increase in the level of centrosomal Arp2/3 was also observed when we analysed the association of p34 with centrosomes purified from cells before and shortly after forced mitotic exit (Fig 4C and D). Additionally, we observed a pool of Arp2 colocalised with centrin at the centre of centrosomes, which was significantly higher during anaphase (Fig EV7A and B′). These data identify a pool of Arp2/3 that is recruited to centrosomes soon after the onset of anaphase, where it functions to nucleate the formation of local actin filaments.

To validate these findings and to determine whether centrosomes might be a potential source of the cytoplasmic pool of actin formed at anaphase, we performed biochemical experiments on centrosomes isolated from cells arrested in prometaphase and those

**Figure 2.  Actin dynamics during forced mitotic exit.**

A   Stills from time lapse of HeLa cells expressing GFP-alpha-tubulin and RFP-Lifeact arrested at prometaphase with STLC (t = 0) and forced to exit mitosis with Cdk1 inhibition (RO-3306) imaged every 90 s. Scale bar = 10 μm and for zoom = 4 μm. n = 21 cells from four independent experiments.

B   Quantification of actin around centrosomes when cells are forced to exit mitosis using Cdk1 inhibition as in (A), showing a similar accumulation of actin as observed in bipolar divisions. Graph shows mean actin accumulation normalised to t = 0, and errors bars indicate standard deviation.

C   HeLa cells expressing GFP-centrin 1 arrested at prometaphase or forced mitotic exit (RO-3306 5′) and stained with phalloidin. Scale bar = 5 μm and for zoom = 2 μm.

D   The level of actin around the centrosome in (C) was quantified and normalised relative to metaphase and shows an increase during forced mitotic exit. STLC arrest 1 ± 0.01317, n = 210; STLC+RO-5 min = 1.251 ± 0.02441, n = 144; unpaired t-test with Welch correction, ****P < 0.0001.

E   Quantification of tubulin intensity around centrosomes during prometaphase-arrested cells or cells forced to exit with Cdk1 inhibitor. Tubulin intensity decreases during forced mitotic exit. STLC-DMSO = 1 ± 0.02407, n = 180; STLC-RO-3306 = 0.6516 ± 0.02337, n = 136. P < 0.0001, Welch's t-test.

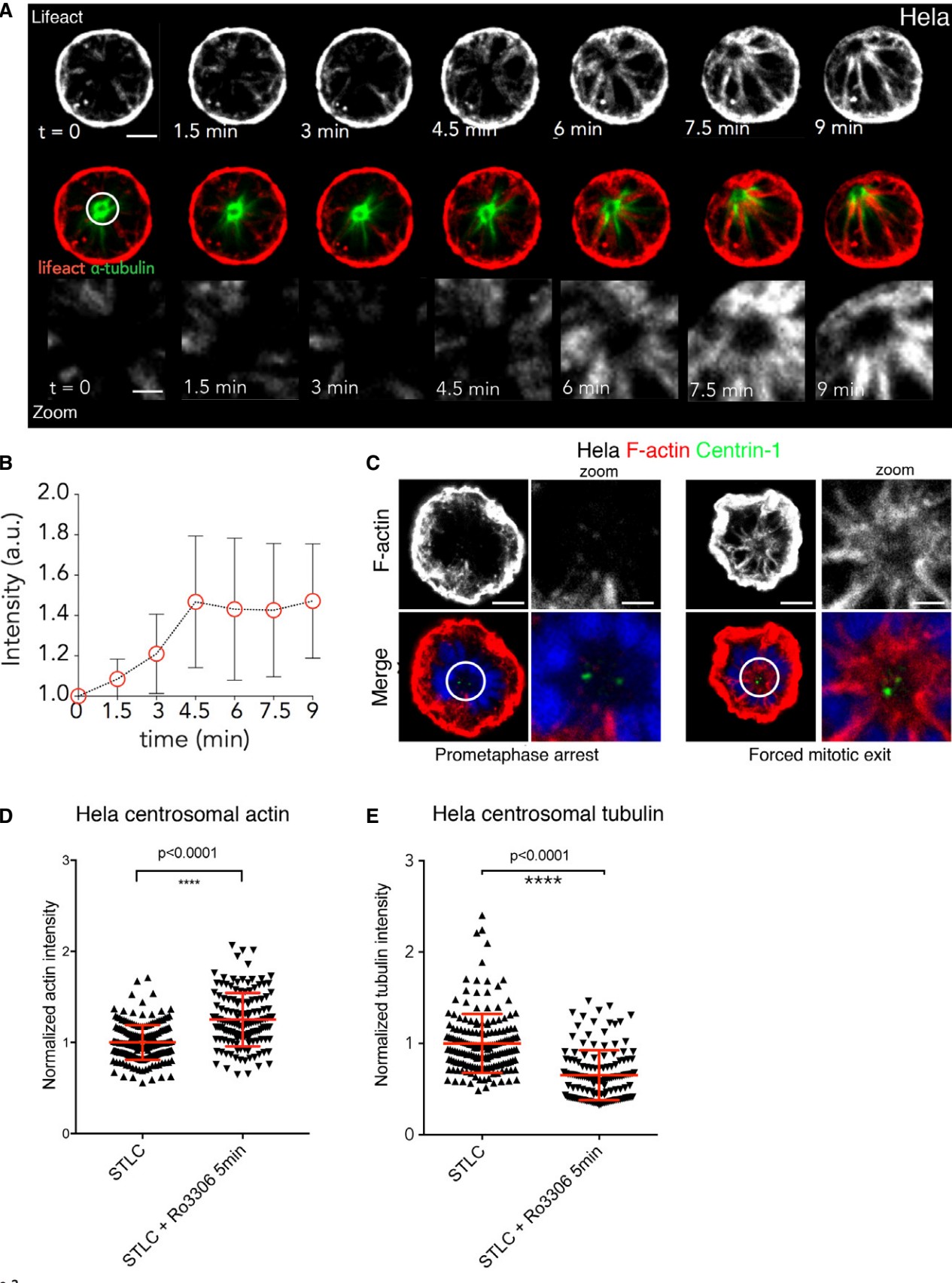

**Figure 2.**

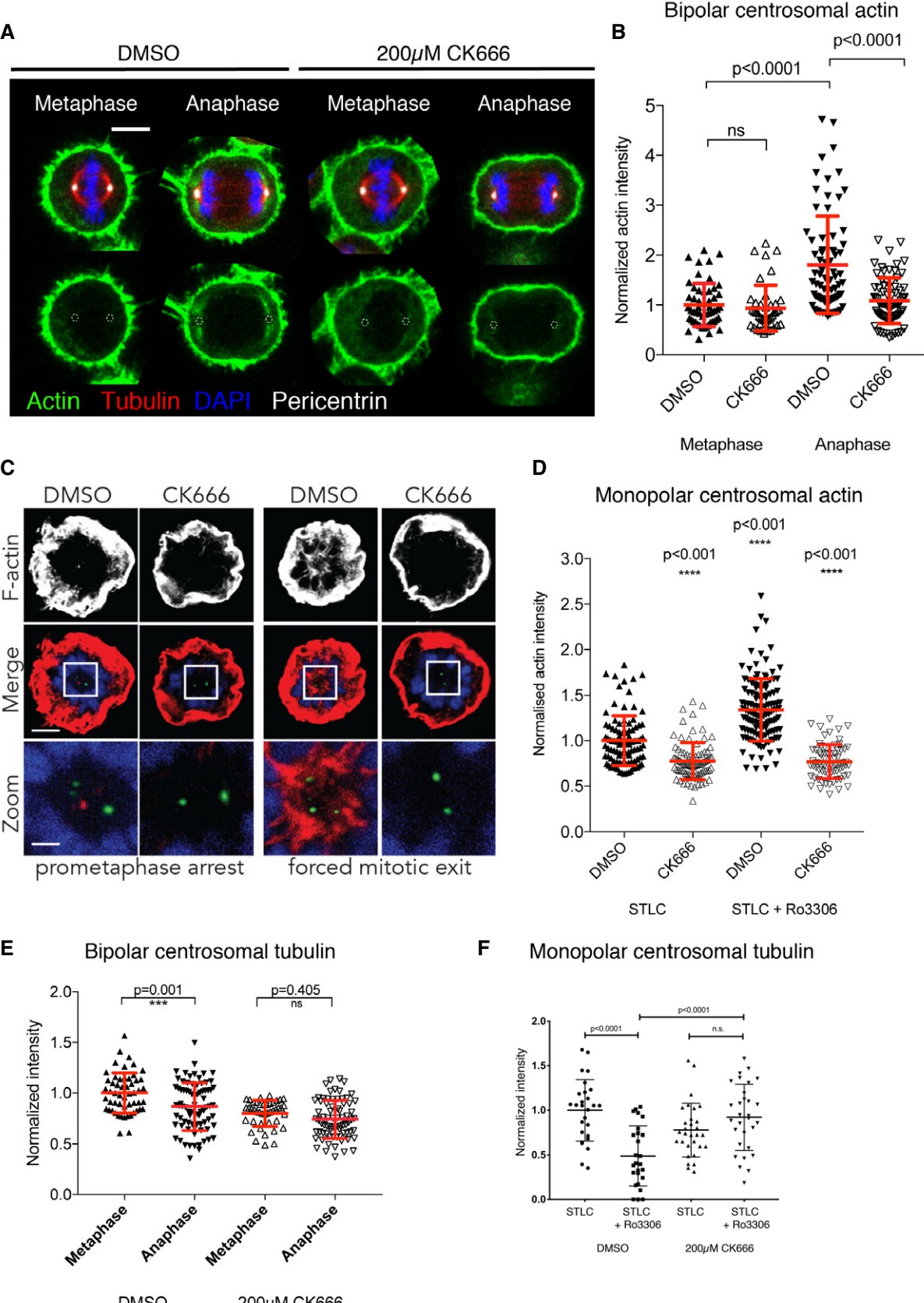

**Figure 3.**

**Figure 3.** Arp2/3-dependent actin accumulation at the centrosome.

A  Maximum projection (2 *z*-slices) view of HeLa cells pre-treated with DMSO and 0.2 mM CK666 for 15 min during their mitotic exit showing that treatment with CK666 leads to reduced accumulation of actin around the centrosomes during anaphase. Scale bar = 10 μm.

B  Quantification of actin around centrosomes for cells treated with DMSO or 0.2 mM CK666, showing the reduction in actin accumulation around the centrosomes following CK666 treatment. DMSO-metaphase = 1 ± 0.4325, *n* = 54; CK666-metaphase = 0.936 ± 0.4604, *n* = 43; DMSO-anaphase = 1.8 ± 0.9736, *n* = 76; CK666-anaphase = 1.087 ± 0.4597, *n* = 73; one-way ANOVA, *P* < 0.0001. Date pooled from three independent experiments.

C  *Z*-projection of HeLa cells expressing GFP-centrin 1 pre-treated with DMSO or 0.2 mM CK666 during prometaphase arrest and forced mitotic exit and stained with phalloidin for F-actin.

D  Quantification of the level of actin around the centrosome from (C), which shows the reduction in actin accumulation around centrosomes following CK666 pre-treatment. DMSO-STLC = 1 ± 0.02768, *n* = 99; CK666 STLC = 0.776 ± 0.02186, *n* = 87; DMSO-RO-3306 = 1.339 ± 0.03048, *n* = 127; CK666-RO-3306 = 0.7699 ± 0.02246, *n* = 70; one-way ANOVA, *P* < 0.0001.

E  Quantification of tubulin around centrosomes for cells treated with DMSO or 0.2 mM CK666, showing the failure to reduce tubulin density around centrosomes following CK666 treatment during bipolar exit. DMSO-metaphase = 1 ± 0.1968, *n* = 54; DMSO-anaphase = 0.867 ± 0.2345, *n* = 78; CK666-metaphase = 0.7995 ± 0.1275, *n* = 43; CK666-anaphase = 0.7407 ± 0.1859, *n* = 74; one-way ANOVA, *P* < 0.0001. Data pooled from three independent experiments. Error bars indicated standard deviation.

F  Quantification of tubulin around centrosomes for cells treated with DMSO or 0.2 mM CK666, during monopolar exit, showing the failure to reduce tubulin density around centrosomes following CK666 treatment. Data from three independent experiments. Prometaphase arrest (p.a.) 1 ± 0.34, *n* = 27; forced mitotic exit (f.m.e) 0.78 ± 0.30, *n* = 26; p.a. plus CK666 0.49 ± 0.34, *n* = 30; f.m.e plus CK666 0.92 ± 0.37, *n* = 30. One-way ANOVA with Sidak's multiple comparisons test.

treated with the Cdk1 inhibitor (see methods and Farina *et al*, 2016). While centrosomes isolated from prometaphase cells failed to nucleate significant levels of actin, centrosomes isolated from cells shortly after forced exit from mitosis nucleated large actin asters (Fig 4E and F). Significantly, the *in vitro* growth of these actin asters could be inhibited by the addition of capping protein (Fig 4G and H), which caps the growing plus ends of filaments (Pollard & Borisy, 2003), as expected if they were formed as the result of active Arp2/3 localised at the centrosome. Further, when we washed-out capping protein and switched the colour of the labelled monomeric actin in the solution, we were able to show that this actin was nucleated at the centre of the aster at the centrosome (yellow dot in Fig 4G). Finally, we used CK666 to confirm that the formation of these actin asters was dependent on Arp2/3 complex activity (Fig 4I and J), as it was in cells exiting mitosis.

Next, we turned to WASH to determine whether this anaphase pool of centrosomal actin filament formation depends on the WASH complex, as was previously described for interphase cells (Farina *et al*, 2016). To begin, we used western blotting to follow WASH1 in interphase cells, mitotic cells or cells following forced mitotic exit (Fig 5A and B). This revealed a clear CDK1-dependent band shift, in line with the idea that WASH is modified by phosphorylation (Olsen *et al*, 2010). In addition, we were able to see a shift in the size of the WASH1 complex on a native gel, during monopolar cytokinesis, suggesting the possibility that there are larger changes in the WASH complex at the transition between metaphase and anaphase (Fig 5B). Finally, we observed a slight elevation of WASH1 signal around centrosomes when we imaged WASH1 localisation in prometaphase-arrested cells and in cells forced to exit mitosis (Fig EV7C).

To test whether WASH complex plays a function in generating the burst of centrosomal actin filament formation at anaphase, we performed WASH1 RNAi (confirmed by Western blot Fig 6C). For this experiment, WASH1 RNAi cells were arrested in STLC and then released (Fig 6A and B) to avoid the impact of WASH RNAi on mitotic entry. When the levels of centrosomal actin were analysed in WASH RNAi cells as they exited mitosis, we observed a near complete loss of centrosomal actin in cells exiting mitosis, both in live (Fig 6A and B) and in fixed samples (Fig 6D and E). Moreover, we observed a similar loss in the ability of centrosomes to nucleate actin asters when we pre-treated purified centrosomes with anti-WASH1 antibody (Fig 6F and G). Thus, both *in vivo* and *in vitro*,

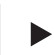

**Figure 4.** Arp2/3-mediated actin nucleation around centrosomes *in vitro*.

A  *Z*-projection of HeLa cells expressing GFP-centrin 1 immunostained for p34-Arc (sub-domain Arp2/3) following prometaphase arrest and forced mitotic exit for 5 min. *N* = 3 independent experiments. Scale bar = 10 μm and for zoom = 2 μm.

B  Levels of p34-Arc around the centrosome (3um area) were quantified relative to metaphase and increase during forced mitotic exit. STLC arrested = 1 ± 0.01576, *n* = 153; STLC+RO-3306 = 1.268 ± 0.01748, *n* = 191; ****P* < 0.0001, unpaired *t*-test with Welch correction. Error bars represent standard deviation.

C  *Z*-projection of centrosomes isolated from cells expressing GFP-centrin 1 during prometaphase arrest and forced exit, which were immunostained with p34-Arc. *N* = 2 independent experiments. Scale bar = 2 μm.

D  Quantification of p34-Arc levels around the centrosome in (C), relative to metaphase, showing its increase during forced exit. STLC arrested = 1 ± 0.00779, *n* = 93; STLC+RO-3306 = 1.249 ± 0.02446, *n* = 134; ****P* < 0.0001, unpaired *t*-test with Welch correction. Error bars represent standard deviation.

E  Time lapse of *in vitro* assay on centrosomes isolated from prometaphase-arrested cell and cells forced to exit mitosis showing centrosomal actin nucleation over time. Scale bar = 10 μm

F  Quantification of actin nucleation from isolated centrosomes as in (E) over an area of 2 μm (white circle), showing an increase in actin nucleation around centrosomes isolated from cells undergoing forced mitotic exit. *N* = 2 isolations, five independent experiments, *n* = 16 STLC and *n* = 19 STLC-RO.

G  Colour switch experiment using green- and red-labelled actin. Red actin was used first, followed by capping protein, and then green-labelled actin. The accumulation of green in the centre indicates that actin is nucleated at the centrosome. Scale bar = 10 μm.

H  Quantification of (G) over the time. *N* = 2 isolations, two independent experiments, *n* = 19 STLC and *n* = 14 STLC-RO. Error bars represent standard deviation.

I  Time lapse showing cytoplasmic actin filament formation at centrosomes isolated from cells undergoing forced exit when pre-treated with DMSO or CK666. Scale bar = 10 μm.

J  Quantification of (I), showing that centrosomal actin nucleation fails when centrosomes isolated from cells undergoing exit are pre-treated with CK666. *N* = 2 isolations, two independent experiments, *n* = 17 RO+DMSO, 26 RO+CK666 24 STLC+DMSO, 23 STLC+CK666. Error bars represent standard deviation.

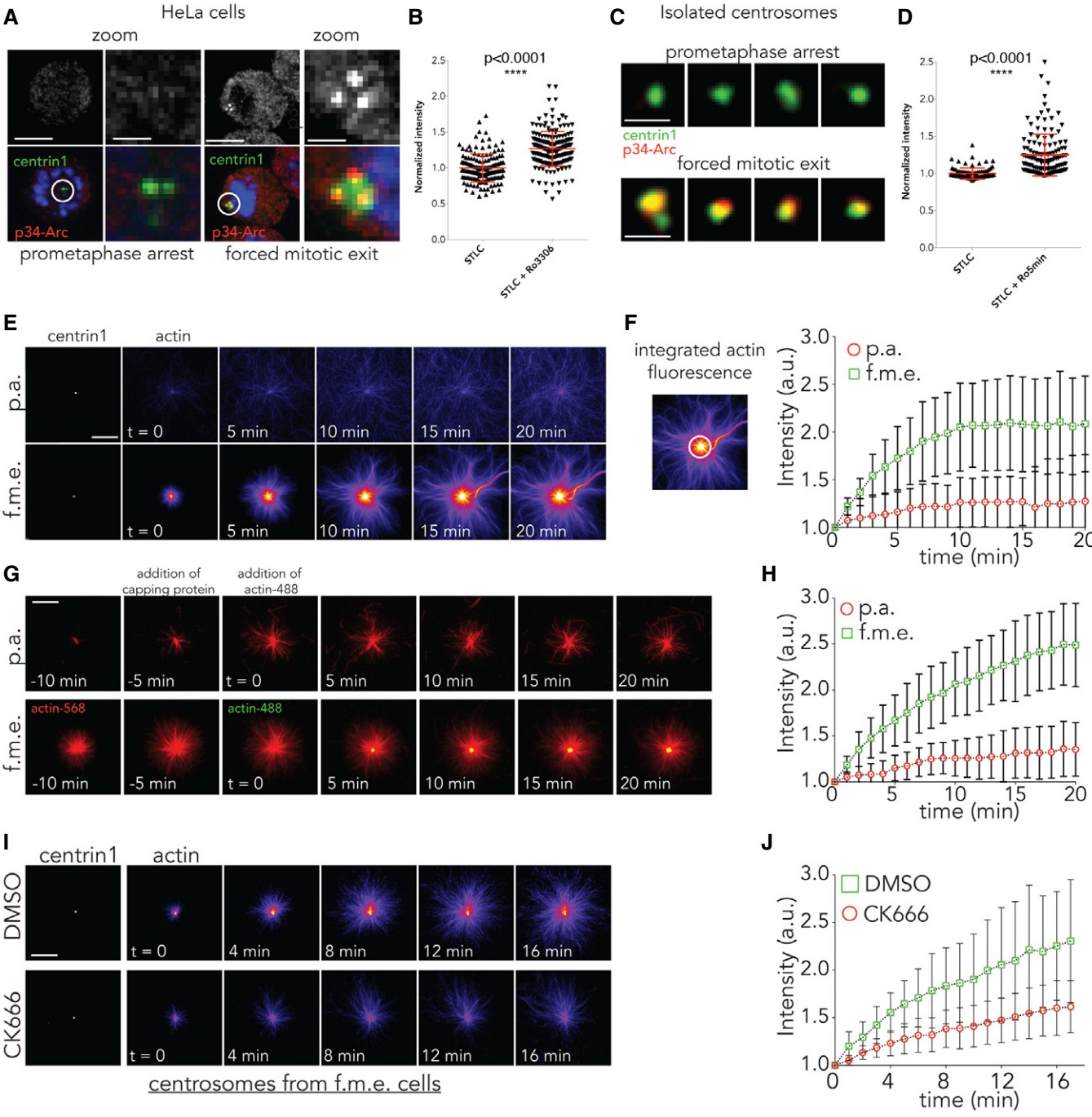

**Figure 4.**

WASH RNAi appears to have the same effect as a treatment with the Arp2/3 inhibitor, CK666.

## Discussion

This paper identifies a role for Arp2/3 and its upstream activator, WASH complex, in the nucleation of actin filaments from centrosomes at mitotic exit. While it has long been clear that the spindle directs the assembly of a contractile actomyosin ring at anaphase, and that actin and microtubules work together to control cell shape and organisation (Huber *et al*, 2015), there has been little evidence of crosstalk between the two filament systems occurring in the opposite direction during mitosis. The data presented here, along with data in studies carried out in lymphocytes in interphase (Inoue *et al*, 2019) and in *Xenopus* egg extracts *in vitro* (Colin *et al*, 2018), suggest that actin may also play a role in tuning microtubule nucleation at the centrosome.

**A**

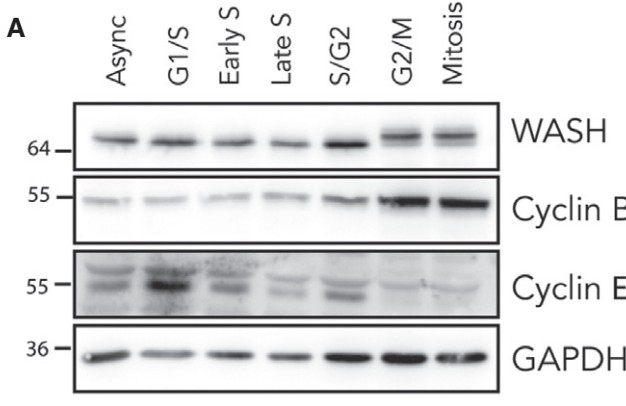

**B**

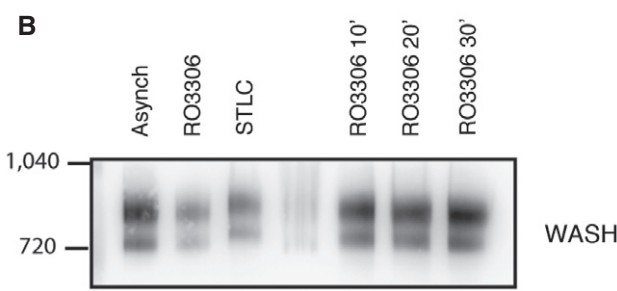

**Figure 5. Localisation and post-translational modifications of WASH1 during mitosis.**

A   Phos-tag Western blot of MCF7-Her2-18 cells synchronised into distinct phases of the cell cycle showing the phospho-shift in WASH1 during mitosis.

B   Blue native PAGE Western blot showing shift in the size of WASH1 protein complex isolated from STLC-treated prometaphase-arrested cells when compared to cells forced to exit mitosis following RO-3306 treatment.

*et al*, 2002; Brangwynne *et al*, 2006; Fakhri *et al*, 2014; Huber *et al*, 2015; Robison *et al*, 2016; Katrukha *et al*, 2017; Colin *et al*, 2018). These interactions occur along the length of microtubules and at their growing plus ends (Akhmanova & Steinmetz, 2015; Mohan & John, 2015), and are especially prevalent at the cell periphery (Waterman-Storer & Salmon, 1997; Wittmann *et al*, 2003), where the two filament systems converge in a crowded space.

The presence of actin at the centrosome (Farina *et al*, 2016) now provides an additional spatial region where the two filaments systems can interact to regulate one another's behaviour. Here, we show that centrosomal actin is generated through the local recruitment and utilisation of the Arp2/3 complex during anaphase. This is likely to be regulated by the change in Cdk1 activity at mitotic exit. The anaphase burst of centrosomal actin is also dependent on the activity of the WASH complex. The timing of centrosomal actin nucleation correlates with the shift in the WASH1 band (Fig 5A and B), suggesting that this process might be regulated by the change in mitotic kinases and phosphatases activity at mitotic exit. This could be augmented by the mitotic regulation of Arp2/3 complex (Figs 4A–D, EV7A and B). The precise nature of the activation of these complexes and their crosstalk remains to be determined.

Like Arp2/3 on a bead, bacterium or patch (Pollard *et al*, 2000; Pollard & Borisy, 2003; Reymann *et al*, 2011), the local activation of Arp2/3 is expected to generate actin filaments that have their minus ends fixed in place, while their plus ends grow out into the cytoplasm (Figs 1–4 and EV1–EV3). This is in keeping with the fact that the extension of these asters can be capped by capping protein (Fig 4G). However, in cells exiting mitosis normally (Figs 1C and D, and EV2, Movies EV1–EV3), centrosomes also appear to nucleate a much more diffuse cytoplasmic pool of actin. This may result from Arp2/3-dependent branching of centrosomally nucleated filaments in the cytoplasm of anaphase cells. Together, this pool of cytoplasmic WASH-Arp2/3-dependent actin may have an impact on the viscosity of the cytoplasm (Moulding *et al*, 2012), as has been shown in meiotic cells (Chaigne *et al*, 2016). Cytoplasmic actin could act in this manner, to limit the movement of separated organelles after the disassembly of the anaphase spindle—as has been shown for mitochondria (Rohn *et al*, 2014).

Previous studies have suggested mitotic roles for cytoplasmic actin in centrosome separation (Rosenblatt *et al*, 2004; Cao *et al*, 2010), pole splitting (Sabino *et al*, 2015) and spindle assembly (Woolner *et al*, 2008; Po'uha & Kavallaris, 2015; Vilmos *et al*,

There are many ways in which the microtubule network can be influenced by actin filaments (Rodriguez *et al*, 2003; Coles & Bradke, 2015; Huber *et al*, 2015; Colin *et al*, 2018). Actin can direct the growth and alignment of microtubules (Kaverina *et al*, 1998; Thery *et al*, 2006; López *et al*, 2014; Elie *et al*, 2015), can change MT dynamics (Zhou *et al*, 2002; Hutchins & Wray, 2014) and can subject MTs to mechanical forces and physical constraints (Gupton

**Figure 6. Anaphase centrosomal actin aster formation depends on the Arp2/3 activator WASH1.**                                                     ▶

A   Time lapse of HeLa cells expressing GFP-alpha-tubulin and RFP-Lifeact treated with control siRNA or siRNA against WASH1 for 48 h arrested at prometaphase and forced to exit mitosis with RO-3306 addition (*t* = 0). Scale bar = 5 µm.

B   Quantification of actin around the centrosome in (A), showing the failure to accumulate actin around the centrosome in siWASH. *N* = 4 experiments. Error bars represent standard deviation.

C   Western blots of lysates from cells treated with siRNA control and siRNA against WASH1 probed with WASH1 antibody, showing the reduction in WASH levels following treatment of cells with siRNA against WASH. Scale bar = 5 µm and 2 µm in zoom.

D   *Z*-projection of HeLa cells expressing GFP-centrin 1 treated with control siRNA or siRNA against WASH, arrested at prometaphase or forced to exit mitosis and immunostained with phalloidin. Scale bar = 5µm

E   Quantification of actin around the centrosome in (F), showing the failure to accumulated actin around the centrosome during forced exit in cells treated with siRNA against WASH. siControl-STLC = 1 ± 0.0189, *n* = 165; siWASH-STLC = 0.9138 ± 0.0298, *n* = 118; siControl-STLC+RO-3306 = 1.911 ± 0.06221, *n* = 147; siWASH-STLC+RO-3306 = 1.103 ± 0.0301, *n* = 140. Error bars represent standard deviation. *P* < 0.0001, one-way ANOVA.

F   Time lapse from *in vitro* assay with centrosomes isolated from cells undergoing forced exit was pre-treated with either no antibody or anti-WASH1 antibody for 1 h. Scale bar = 10 µm.

G   Quantification of *in vitro* assay from (F), showing the reduction in actin nucleation around centrosomes when they are pre-treated with anti-WASH1. Error bars represent standard deviation. Two independent experiments.

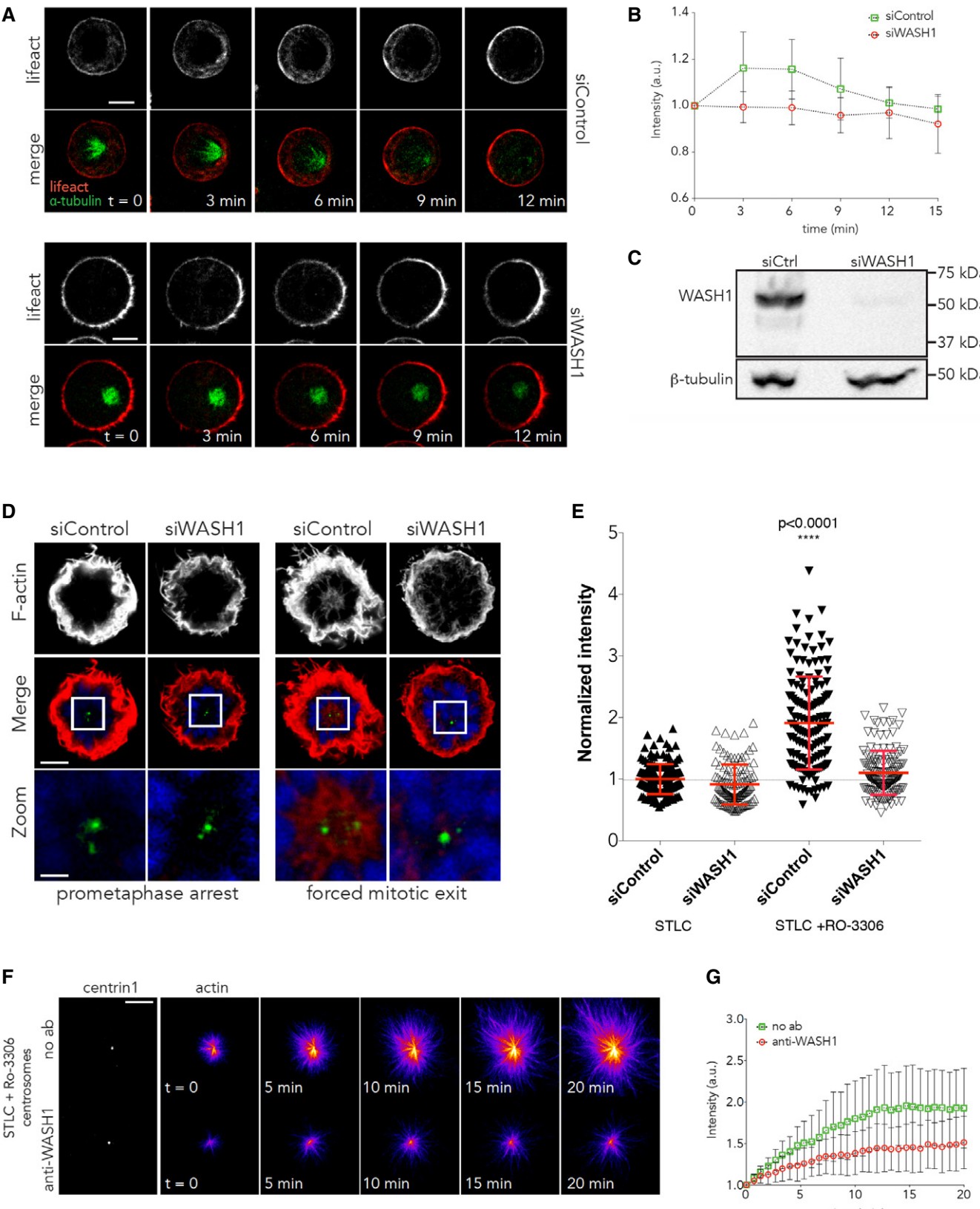

Figure 6.

2016). Our data suggest that the capacity for the generation of centrosomal actin in metaphase is limited. Moreover, in HeLa cells, where this cytoplasmic actin was first observed (Mitsushima *et al*, 2010; Fink *et al*, 2011), Arp2/3 complex activity has little influence on the timing of mitotic progression (Lancaster *et al*, 2013) and contributes little to metaphase cortical mechanics (Chugh *et al*, 2017). However, our data support the idea that cytoplasmic actin has a significant function at the onset of anaphase (including in the control of mitochondrial movement (Rohn *et al*, 2014)), when CDK1-mediated phosphorylation of WASH is relieved and active Arp2/3 returns to the centrosome to induce a burst of actin.

This burst in centrosomal actin is accompanied by a reduction in the microtubule density around the centrosomes at early anaphase. In the absence of centrosomal actin nucleation, with Arp2/3 inhibitor, this reduction in microtubule density is inhibited. This suggests an antagonistic crosstalk between the two filament systems at mitotic exit. The precise nature of this antagonism between local actin and microtubule formation at the centrosome remains to be determined (Piel *et al*, 2001; Obino *et al*, 2016; Inoue *et al*, 2019). We speculate that this burst of anaphase actin nucleation at centrosomes may cause steric problems, as the two systems compete for physical space around the centrosome. However, in live cells, we cannot rule out the possibility of an indirect crosstalk mediated by either changes in localisation or activity of proteins affecting post-translational modifications of tubulin during mitotic exit (Shi *et al*, 2019). Further, it is possible that the centrosomal actin pool could have a more profound effect depending on the nature of microtubules, i.e. astral vs spindle microtubules at mitotic exit.

The actin dependent reduction in microtubule density around centrosomes may help cells leaving mitosis to re-establish a telophase array of sparse, long centrosomal microtubules. These microtubules can then span the entire space of the cells, assisting in polar relaxation, adjusting spindle orientation (Kiyomitsu & Cheeseman, 2013) and/or promoting re-spreading (Ferreira *et al*, 2013). In short, this reduction in microtubule density at early anaphase could assist cells exiting mitosis to return to their interphase organisation and morphology.

# Materials and Methods

### Cell culture

HeLa Kyoto cells were cultured in DMEM, and Jurkat cells (immortalised human T lymphocytes) as well as MAVER1 CML B cells in RPMI 1640 (Gibco) at 37°C and 5% $CO_2$. All media were supplemented with 10% foetal bovine serum and penicillin/streptomycin (Gibco). Cells were synchronised in prometaphase using S-trityl-L-cysteine (STLC; Sigma) at 5 μM treatment for 18 h (Skoufias *et al*, 2006). Forced mitotic exit was performed by the addition of 20 μM RO-3306 (Enzo Life Sciences), an inhibitor of Cdk1/cyclin B1 and Cdk1/cyclin A to STLC-treated cells. For the Arp2/3 complex inhibition experiments, we used 0.2 mM CK666 (Sigma-Aldrich). Control experiments were performed using DMSO. For CK666 treatment (Fig 3), cells were plated with 2 mM thymidine for 22 h, were released from thymidine block for 9 h, following which they were treated with DMSO or 0.2 mM CK666 for 15 min. They were then processed for staining. Actin network disruption was performed by adding 10 μg/ml cytochalasin D (Sigma-Aldrich). Microtubule

depolymerisation was performed by adding 1 μM nocodazole (Sigma-Aldrich) for 1 h at 37°C and 5% $CO_2$ and for 30 min at 4°C.

### Stable and transient cell transfection

Stable HeLa cell lines expressing GFP-α-tubulin and RFP-Lifeact were established starting from HeLa stably expressing GFP-α-tubulin by lentiviral transduction with rLV-Lifeact-RFP (Ibidi) using 3 MOI (multiplicity of infection). After 24 h, cells were incubated with fresh medium for 48 h. After 72 h post-transduction, stable cells were selected using 1 μg/ml puromycin. Medium with puromycin was replaced every 2–3 days until resistant colonies were identified. ON-TARGETplus Human SMARTpool WASH1-targeting (siWASH1) siRNAs (Dharmacon, GE Healthcare) were transfected into HeLa cells at a final concentration of 20 nM using Lipofectamine RNAiMax (Life Technologies) according to supplier's protocol. Negative control siRNA was performed using AllStars Negative Control siRNA (Qiagen).

### Isolation of centrosomes

Centrosomes were isolated from cells arrested in prometaphase (STLC) or shortly after their forced mitotic exit (RO-3306 5′) using a previously published protocol (Farina *et al*, 2016). Cells were incubated for 18 h with STLC and then treated with nocodazole (0.2 μM) and cytochalasin D (1 μg/ml). For the mitotic exit, we added to STLC-treated cells 20 μM RO-3306 for 5 min, and cells were kept on ice for 30 min. Centrosomes were then harvested by centrifugation onto a 60% sucrose cushion and further purified by centrifugation through a discontinuous (70, 50 and 40%) sucrose gradient. Composition of sucrose solutions was based on TicTac buffer (10 mM Hepes, 16 mM Pipes (pH 6.8), 50 mM KCl, 5 mM $MgCl_2$, 1 mM EGTA). The TicTac buffer was supplemented with 0.1% Triton X-100 and 0.1% β-mercaptoethanol.

### Protein expression and purification

Tubulin was purified from fresh bovine brain by three cycles of temperature-dependent assembly/disassembly in Brinkley Buffer 80 (BRB80 buffer: 80 mM Pipes pH 6.8, 1 mM EGTA and 1 mM $MgCl_2$) according to Shelanski (Shelanski, 1973). Fluorescent tubulin (ATTO-565-labelled tubulin) was prepared according to Hyman *et al* (1991). Actin was purified from rabbit skeletal-muscle acetone powder (Spudich & Watt, 1971). Monomeric Ca-ATP-actin was purified by gel-filtration chromatography on Sephacryl S-300 (MacLean-Fletcher & Pollard, 1980) at 4°C in G buffer (2 mM Tris–HCl, pH 8.0, 0.2 mM ATP, 0.1 mM $CaCl_2$, 1 mM $NaN_3$ and 0.5 mM dithiothreitol (DTT)). Actin was labelled on lysines with Alexa-488, Alexa-568 and Alexa-647 as described previously (Isambert *et al*, 1995; Egile *et al*, 1999). Recombinant human profilin, mouse capping protein, the Arp2/3 complex, GST-pWA and mDia1 were purified according to previous work (Almo *et al*, 1994; Egile *et al*, 1999; Machesky *et al*, 1999; Falck *et al*, 2004; Michelot *et al*, 2007).

### *In vitro* assay

This was done essentially as in Farina *et al* (2016). Briefly, experiments were performed in polydimethylsiloxane (PDMS) open chambers in order to sequentially add experimental solutions when

needed. PDMS (Sylgard 184 Kit, Dow Corning) was mixed with the curing agent (10:1 ratio), degased, poured into a Petri dish to a thickness of 5 mm and cured for 30 min at 100°C on a hot plate. PDMS layer was cut to 15 × 15 mm and punched using a hole puncher (ted Pella) with an outer diameter of 8 mm. The PDMS chamber and clean coverslip (20 × 20 mm) were oxidised in an oxygen plasma cleaner for 20 s at 80 W (Femto, Diener Electronic) and brought into contact. Isolated centrosomes were diluted in TicTac buffer (10 mM Hepes, 16 mM Pipes (pH 6.8), 50 mM KCl, 5 mM $MgCl_2$, 1 mM EGTA) and incubated for 20 min. Excess centrosomes were removed by rinsing the open chamber with large volume of TicTac buffer supplemented with 1% BSA to prevent the non-specific interactions (TicTac-BSA buffer). Microtubules and actin assembly at the centrosome were induced by diluting tubulin dimers (labelled with ATTO-565, 30 μM final) and/or actin monomers (labelled with Alexa-488, or Alexa-568, or Alexa-647, 1 μM final) in TicTac buffer supplemented with 1 mM GTP and 2.7 mM ATP, 10 mM DTT, 20 μg/ml catalase, 3 mg/ml glucose, 100 μg/ml glucose oxidase and 0.25% w/v methylcellulose. In addition, a threefold molar equivalent of profilin to actin was added in the reaction mixture. Antibody inhibition experiments were performed by incubating isolated centrosomes with primary antibodies (diluted in TicTac-BSA buffer) for 1 h. The control experiment without antibodies was performed incubating isolated centrosomes for 1 h with TicTac-BSA buffer. Arp2/3 complex inhibition experiments were performed by adding 0.2 mM CK666 in the reaction mixture.

## Immunofluorescence staining (in cell)

This was done essentially as in Farina *et al* (2016). Briefly, cells were incubated for 18 h with STLC. Forced mitotic exit was performed incubating STLC cells with RO-3306 for 5 min. Cells were then fixed and stained. For actin filament staining, cells were fixed with 4% paraformaldehyde (PFA) for 20 min, blocked with antibody blocking buffer (PBS supplemented with 1% BSA, PBS-BSA) for 30 min. Permeabilisation was performed with 0.2% Triton X-100 for 1 min. Alexa-647-phalloidin (200 nM) was incubated for 20 min. DNA was labelled with a 0.2 μg/ml solution of 4′,6-diamidino-2-phenylindole dihydrochloride (DAPI) (Sigma). The coverslips were air-dried and mounted onto glass slides using Mowiol mounting medium. Arp2/3 staining was performed by fixing cells with methanol at −20°C for 3 min and blocking with PBS-BSA for 30 min. Primary and secondary antibodies, diluted in PBS-BSA, were incubated for 1 h and 30 min, respectively. DNA labelling and coverslip mounting were performed as previously described.

## Immunofluorescence staining (isolated centrosomes)

This was done essentially as in Farina *et al* (2016). Briefly, staining of F-actin on centrosomes was performed without prior fixation. Isolated centrosomes were incubated with primary antibodies for 1 h and with secondary antibodies for 30 min at room temperature. The antibodies were diluted in TicTac-BSA buffer.

## Imaging, processing and analysis

This was done essentially as in Farina *et al* (2016). Briefly, fixed cell images were captured on a confocal microscope (Leica SP5) using a 40× 1.25 N.A. objective lens, 63× 1.4 N.A. or Zeiss LSM800 with a 63× 1.4N.A. lens. Live cell imaging was performed on a UltraView Vox (Perkin Elmer) spinning disc confocal microscope with 60× NA 1.4 oil objective and 100× 1.4 N.A. and 3I spinning disc confocal with 63× 1.4 N.A and 100× 1.4 N.A objectives equipped with a temperature-controlled environment chamber. Image processing was performed using ImageJ software. All the images show the centrosome plane. Measurement of the actin amount around the centrosome was performed by measuring the integrated intensity of fluorescence in a 4-μm-diameter circle centred around the centrosome. p34-Arc measurements were performed measuring the integrated fluorescence intensity in a 3-μm-diameter circle centred around the centrosome. Data from separate experiments were normalised so that the average intensity in control cells was 1. Imaging of isolated centrosomes was performed with a total internal reflection fluorescence (TIRF) microscope (Roper Scientific) equipped by an iLasPulsed system and an Evolve camera (EMCCD 512 × 512, pixel = 16 μm) using a 60× 1.49 N.A objective lens. Actin nucleation activity was quantified measuring the actin fluorescence intensity integrated over a 2 μm diameter at the centre of the actin aster and normalised with respect to initial intensity over the time. Representative data for several experiments are shown.

## Actin staining and measurement bipolar divisions (Figs 1 and 3)

Fixed cells—Cells were cultured in 96-well cell carrier plates. For staining, cells were fixed with 4% paraformaldehyde (PFA) for 20 min and permeabilised with 0.1% Triton X-100 for 10 min. They were blocked in antibody blocking buffer (PBS supplemented with 5% BSA, PBS-BSA) for 30 min and incubated with primary antibodies anti-tubulin (1:400) and anti-pericentrin (1:1,000) overnight at 4C. Cells were incubated with secondary antibodies—goat anti-mouse Alexa-647, goat anti-rabbit Alexa-568, phalloidin-FITC (1:500) and DAPI (4′,6-diamidino-2-phenylindole dihydrochloride-1:1,000) for 1 h at room temperature. Jurkat cells were fixed with 4% paraformaldehyde (PFA) for 20 min in suspension and were adhered to 96-well plates coated with Poly-L-lysine.

For analysis, a 4-μm-diameter (for HeLa) and 3-μm-diameter (for Jurkat) circle was centred around the centrosome using the pericentrin channel and the corresponding actin- and tubulin-integrated density around the same region was measured using Fiji. The images used were 2 *z*-projection around the centrosome plane. In order to combine data from multiple experiments, the data were normalised to metaphase average for each cell line. For live cells, a 4-μm-diameter circle was centred around centrosomal region using alpha-tubulin RFP as reference. A 4 *z*-projection around centrosome region was used for measurements. The intensity data for each cell were normalised to the intensity at *t* = 0, which is one frame before anaphase onset.

## Actin measurement forced exit (Figs 2E and F and EV1)

A circular, 3 μm$^2$ ROI was centred on the centrosomes as determined by pericentrin staining. A series of slices equal to approximately 3 μm in height (with z interval of 0.39 μm, the number of slices was 9, including the central slice) were then analysed using the defined ROI to give a mean intensity per slice, which were then averaged to give the mean intensity within the 3 × 3 × 3 cylindrical

area. Individual cell averages were normalised to the experimental average for the STLC condition.

## Statistics

Statistical analysis was performed with GraphPad Prism 7 (GraphPad Software). All graphs show mean and error bars are standard deviation. The test used is mentioned for each graph in figure legend. In all the graphs, prism convention: ns ($P > 0.05$), *($P \leq 0.05$), **($P \leq 0.01$), ***($P \leq 0.001$) and ****($P \leq 0.0001$).

## Western blotting

Western blots were performed fractioning proteins on SDS polyacrylamide gels. Membrane blocking was carried out using 3% BSA in PBS. Primary and secondary antibodies were diluted in PBS supplemented with 1% BSA and 0.1% Tween-20 while washing steps were performed with PBS supplemented with 1% BSA and 1% Tween-20.

## Blue NativePAGE

This was done essentially as in Tyrrell *et al* (2016). Briefly, cells were lysed using the NativePAGE™ Sample Prep Kit (Thermo Scientific—BN2008). Each sample was lysed in 400 μl of NP-lysis buffer [100 μl of NP buffer, 260 μl of ddH$_2$O, 40 μl 5% digitonin, halt protease inhibitor (Pearce) and halt phosphatase inhibitor (Pearce)] for 10 min on ice and scraped with a cell scraper. Samples were centrifuged at 20,000 $g$ for 30 min at 4°C, and pellets were discarded. Equal amounts of protein (determined using DC™ Protein Assay Kit (Bio-Rad—5000111)) were separated by BN-PAGE NativePAGE™ Novex™ 3–12% Bis-Tris Protein Gels as per the manufacturer's instructions (NativePAGE™ Bis-Tris Gel protocol from Thermo Scientific). Native-Mark™ Unstained Protein Standard (Thermo Fisher Scientific) was used as molecular weight standard. Protein was blotted onto PVDF membranes and fixed for 15 min in 8% acetic acid. Membranes were blocked in 5% milk for 1 h at room temperature and incubated overnight rocking at 4°C in 5% milk PBS-T. Membranes were washed in PBS-T and incubated for 1 h with peroxidase-conjugated secondary antibody (Cell Signalling, Hitchin, UK). Clarity Western blotting ECL substrate (Bio-Rad) was used to generate a signal that was detected using a Chemidoc imaging system (Bio-Rad).

## Phos-tag band shifts

Cells were lysed using hot lysis buffer (2% SDS, 1 mM EDTA, 50mMHaF, preheated to 97°C). Samples were allowed to cool at RT before performing protein assay using DC™ Protein Assay Kit (Bio-Rad—5000111). Sample buffer was added, and equal protein was loaded into 8% polyacrylamide gels supplemented with 5 μM Phos-tag and 10 μM MnCl$_2$. Phos-tag gels were resolved at 30 mA under constant current until the dye front migrated to the bottom of the gel. Gels were transferred onto nitrocellulose membrane with a 0.45 pore size (GE Healthcare) at 250 mA for 120 min under constant current.

## Antibodies and chemicals

For immunofluorescence staining, we used the following antibodies: mouse anti-p34-Arc (Dubois *et al*, 2005) (undiluted), mouse anti-tubulin (Sigma T9026), rabbit anti-pericentrin (ab4448), phalloidin-FITC (P5282) and Alexa-647 phalloidin (Fluka 65906). For inhibition experiments on isolated centrosomes, we used rabbit anti-WASH antibodies (Derivery *et al*, 2009) (2 μg/ml) (gift Alexis).

For Western blot analysis, we used rabbit anti-WASH1 (1: 1,000, ab157592, Abcam), rabbit anti-WASH (1:1,000; Atlas), strumpellin (1: 1,000, ab101222, Abcam) and anti-beta-tubulin (sc-5274, Santa Cruz). Labelled anti-mouse and anti-rabbit secondary antibodies (1: 1,000) and HRP-conjugated goat IgG anti-mouse and anti-rabbit (1: 10,000) for Western blot were obtained from Jackson ImmunoResearch.

Cytochalasin D and nocodazole were purchased from Sigma-Aldrich. PFA was purchased from Delta Microscopies. CK666 was purchased from Sigma-Aldrich. Alexa-647-phalloidin was purchased from Life Technologies.

**Expanded View** for this article is available online.

## Acknowledgements

NR, FF and BB thank CRUK. NR and BB thank BBSRC. LB (Louise) was funded by a Breast Cancer Now grant (2014MayPR292) to TZ. TW is funded by the NLD BBSRC doctoral training programme (BB/M011186/1/1797330). TZ thanks the Institute of Translational Medicine Biomedical Imaging Facility. LB (Laurent) is supported by ERC grant (AAA 741773). GS thanks AIRC (IG#18621) and Italian Ministry of Health (RF-2013-02358446).

## Author contributions

NR carried out the cell culture experiments for normal bipolar cytokinesis in different cell lines, live imaging bipolar and monopolar cytokinesis. FF carried out the *in vitro* work and cell biological experiments shown in Figs 2, 3C, 4 and 6, Supp 5). DS-E and JA helped with monopolar and bipolar cytokinesis experiments, respectively. FF and BB conceived the initial idea. NR and BB oversaw the development of the project and wrote the manuscript with assistance from TZ. LB (Louise) and TZ did WASH biochemistry. TW, JB and TZ did the monopolar cytokinesis in Jurkat and MAVER1 cell lines. MT and LB (Laurent) oversaw *in vitro* work and advised on *in vivo* work, together with help from GS.

## Conflict of interest

The authors declare that they have no conflict of interest.

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
