## [Review Process File · The EMBO Journal]

Local actin nucleation tunes centrosomal microtubule nucleation during passage through mitosis

Francesca Farina, Nitya Ramkumar, Louise Brown, Durren Samander-Eweis, Jannis Anstatt, Thomas Waring, Jessica Bithell, Giorgio Scita, Manuel Thery, Laurent Blanchoin, Tobias Zech, Buzz Baum.

Review timeline:

Submission date:	16 th June 2018
Editorial Decision:	20 th July 2018
Revision received:	24 th November 2018
Editorial Decision:	10 th January 2019
Author Correspondence:	16 th January 2019
Editor Correspondence:	24 th January 2019
Editorial Decision:	14 th March 2019
Revision received:	2 nd April 2019
Accepted:	4 th April 2019

Editor: Hartmut Vodermaier

Transaction Report:

1st Editorial Decision

20th July 2018

Thank you again for submitting your manuscript on centrosomal actin/microtubule interplay during mitosis for our editorial consideration. We have now received the enclosed reports from three expert referees, who on the whole consider your results potentially interesting but also raise a number of well-taken concerns that would have to be satisfactorily answered prior to publication. In particular, it seems that a majority of referees is not convinced by the data supporting in cellulo relevance of the key findings.

Should you be able to decisively address this key issue as well as the various other specific points, we would be open to considering a revised manuscript further for publication in The EMBO Journal. Nevertheless, given that it is unclear if all major points might be straightforwardly addressable during a regular revision round, I would in this case encourage you to contact me with a tentative response and proposal for how you could might answer the referee concerns already during the early stages of the revision period; so we could discuss which revisions would be essential/reasonable and whether a less-extensively revised revision might alternatively become suited for rapid publication in one of our sister journals such as EMBO Reports.

REFeree REPORTS.

Referee #1:

Based on a previous study nicely showing that the centrosome is a actin nucleation organising center the authors now propose actin nucleation at the centrosome is important to limit or inhibit centrosomal microtubule assembly. They further confirm from the previous reports that Arp2/3 and

WASH are important for this centrosomal actin assembly during mitotic exit. The study is potentially interesting and the in vitro work is beautiful but the evidence for physiological mitosis is somewhat scarce. The criticism regarding this aspect is outlined below.

Figure 1:

The authors write about a "dramatic increase in the levels of actin associated with centrosomes (Figure 1A-C)."

However, one cannot see any distinct actin structures in 1A, other than a actin cloud in the cytosol of cells that have rounded up during mitosis. In addition, the centrosome was not labeled in this imaging analysis. In any case, centrosomal actin cannot be seen at least in the images provided to this reviewer.

In addition, it would be much better and more convincing if the authors would provide similar examples from other cell types (possibly labeled for the centrosome using γ -tubulin or centrin1 etc) or even primary cells rather than a redundant HeLa duplicate example. This is in particular important since HeLa cells do vary a lot between different laboratories and can have substantial cytogenetic defects. Along these lines it would be good to see another actin marker in addition to LifeAct (e.g. F-tractin) so substantiate the claim and to exclude possible artefacts emanating from the probe itself.

Figure 2:

Throughout the manuscript the HeLa example given in Figure 1 remains the only spontaneous physiological mitosis not involving long time drug manipulations such as performed in the other figures.

This STLC-arrested and Ro3306-mediated (Cdk1 inhibition) mitotic exit is a long term treatment. While this is surely a valid way to study some aspects of the cell cycle through drug treatments one would still like to see this phenomenon in a more physiological context.

If the authors would like to make the point that centrosomal actin assembly is critical for proper mitosis they should more convincingly show this in cells undergoing mitosis while adding Arp2/3 inhibitor under the microscope.

In Fig 2B in DMSO control cells one can again only observe a cytoplasmic actin cloud. Efforts to improve imaging resolution may be desirable.

Figure 3C, did the authors test whether other actin assembly factors colocalize with centrin1? Some specificity control would be good to see here. For example, GFP-WASP?

Figure 4A, the proposed "shifting in size of strumpelin" (figure legend 4A) is not very convincing. There seems to be also no reference to strumpelin in the main text.

Did the authors look at Arp2/3 ko cells? Do they display this phenotype?

Referee #2:

Farina et al. present some experiments which indicate that the actin nucleated at centrosomes at the end of mitosis downregulates microtubule nucleation. It's an interesting set of observations at the interface of these two cytoskeletal systems during cell division. The conclusions seem to me to be well supported by the data, taking advantage of in vitro and in cellulo approaches. I have some suggestions for the authors to improve the presentation of their paper (not presented in order of importance).

Correction for multiple comparisons: Experiments with more than two groups cannot be tested using multiple Student's t-tests, this is subject to increased likelihood of Type I error. An ANOVA with a suitable post-hoc test (e.g. Tukey's or Dunnett's with a single control) should be used instead. This applies if the data is normally distributed, please test for this first. This affects the statistical test in Fig 1C, Fig 3L; assuming what is written in the Methods is correct. The test should be declared in

the legend. I'd encourage the authors to put the p-values on the graph rather than the star fest.

In all graphs, markers and error bars should be defined in the legend (what do they show? mean/median {plus minus} SD/SEM/95%CI).

Fig 1B As I understand it this plot shows data from the 2 cells shown in 1A, why are there error bars? N=1 for each line on the graph. I'm guessing it reflects multiple ROIs, multiple z-slices or the two centrosomes. If so it's not correct, bars should be removed.

Figs 2F,2H,3B,3D,4G suffer from too many points and oversized markers. They should make the markers smaller, use transparency or better still add box-whisker or violins to show the shape of the distribution. At the absolute minimum, they should shrink the markers to dots and include an error bar indicating the standard deviation, like the plot in 3L.

Results 2nd para (page & line numbers would've been handy!): this reads a bit strangely. STLC and Cdk1 inhibitor are applied but there's no rationale given. "This changed when cells were forced to undergo..." this sounds strange - almost like the authors are surprised that someone was applied the drug. It would make more sense to say: The motivation is to better visualize centrosomal actin (the isolation part can be described later), so to do this we used STLC to blah blah and then forced exit by blah blah. The paper is generally easy to understand and well written, but this part could be improved. The FME method is used in all the subsequent figures so it feels like it should be better introduced.

Fig 4I legend should read anti-WASH1? The experiments are a depletion of WASH not supplementation.

Referee #3:

This manuscript by Farina and colleague investigates the role of actin at centrosomes during mitotic exit. The authors suggest a model that a burst of Arp2/3-dependent actin nucleation at the centrosome at anaphase aids the disassembly of centrosomal microtubules (MTs). While this is an attractive model, the authors do not present sufficient evidence to support this model. This manuscript, similar to published data on *in vivo* centrosomal actin, suffers from two major problems 1) the low quality of the light microscopy, and 2) the inability to specifically perturb the actin nucleation machinery at centrosomes independent of the entire cell. The authors overinterpret their images, and their global drug treatments are used to strongly conclude specificity to centrosomes. I do not agree with their conclusions given the presented data.

Major comments

Figure 1 shows live analysis and measurements of unperturbed cells. There are two concerns here:

1. The system: The authors attempt to measure actin and MTs at the centrosome, but the images presented are of the spindle pole region. This is due to the notorious HeLa cell rounding, which is why HeLa cells are generally avoided when studying MTs in mitosis. The authors should identify a more suitable system, or greatly improve their imaging. Imaging such as the following example of a HeLa cell is necessary to quantify the MT ring surrounding the centrosomes

<https://www.sciencedaily.com/releases/2010/03/100310091652.htm>

2. The quantification. The authors should perform several control measurements of actin throughout the cell such as the cortex and different regions of the cytoplasm. Additionally, the authors should treat unperturbed (no STLC) cells with CK666 and quantify changes in MTs and actin over time from their live movies.

Figure 2 There are two concerns here as well:

1. The system: To overcome the imaging limitation of HeLa cells, the authors use a two-drug system to isolate centrosomes away from the cortex (STLC) and drive cells out of mitosis (Ro-3306). Thus, the controls used in this study are cancer cells that are not amenable to imaging and have been treated with two drugs. Although, the images in 2A are the most convincing evidence of an *in vivo* centrosome having the capability of nucleating actin, these cells are heavily manipulated, throwing into question the relevance of this observation.

2. The quantification: The authors should present both Actin and MT measurements for these live

experiments. It would also be helpful if the intensity axes are labeled with what is measured.

The most telling statement in the manuscript, and one that concedes that no specificity can be determined using these global drug treatment is "Just as observed in cells treated with CK666, the silencing of WASH by RNAi led to the near complete loss of cytoplasmic actin in cells exiting mitosis, as assayed using both live imaging (Figure 4C-D) and in cells that had been fixed and stained for F-actin (Figure 4F-G)."

Other comments

a. In the introduction, the authors refer to MT dynamics in mitosis and reference Mchedlishvili. This manuscript does not quantify MT dynamics, which is a specific term used to describe events of growing, shrinking, pausing, catastrophe and rescue. Better references to changes in MT dynamics are found from the Wadsworth lab from the early 2000s. Similarly, at the end of the introduction the authors state they have identified a role for actin nucleation in the regulation of mitotic microtubule dynamics. This is not an accurate representation of the work.

b. The following statement is an unsubstantiated conclusion from the previous paper, which was also unable to show specific loss of Arp2.3 at the centrosome - "Since previous work demonstrated that the actin formed at interphase centrosomes was nucleated by a pool of Arp2.3 present at centrosomes (Farina et al., 2016)."

c. The following two relevant statements are conjecture -
"In metaphase cells, this revealed a low level of cytoplasmic actin". The images from Figure 1 show substantial cytoplasmic actin.

"As previously noted, CK666 did not have a profound impact on the formation of the mitotic cortical actin mesh." Is there a specific reason to suspect that only profound changes in actin would result in meaningful changes of cellular physiology? Cortical F-Actin should be measured in untreated and CK666 treated cells.

1st Revision - authors' response

24th November 2018

Referee #1:

Based on a previous study nicely showing that the centrosome is a actin nucleation organising center the authors now propose actin nucleation at the centrosome is important to limit or inhibit centrosomal microtubule assembly. They further confirm from the previous reports that Arp2/3 and WASH are important for this centrosomal actin assembly during mitotic exit. The study is potentially interesting and the in vitro work is beautiful but the evidence for physiological mitosis is somewhat scarce. The criticism regarding this aspect is outlined below.

Figure 1:

The authors write about a "dramatic increase in the levels of actin associated with centrosomes (Figure 1A-C)."

However, one cannot see any distinct actin structures in 1A, other than a actin cloud in the cytosol of cells that have rounded up during mitosis. In addition, the centrosome was not labeled in this imaging analysis. In any case, centrosomal actin cannot be seen at least in the images provided to this reviewer.

We apologise for the poor quality of these images. We have replaced figure 1 with stills from higher resolution movies in which it is clear that there is a transient actin accumulation at centrosomes. In addition, we have included a panel of fixed cells (Hela and Jurkat cells) immunostained for F-actin (Phalloidin), tubulin and pericentrin (for centrosomes) at metaphase and anaphase. Here again with quantification, we show that there is an accumulation of actin and decrease in tubulin at early anaphase compared to cells in metaphase.

In addition, it would be much better and more convincing of the authors would provide similar

examples from other cell types (possibly labeled for the centrosome using γ -tubulin or centrin1 etc) or even primary cells rather than a redundant HeLa duplicate example.

We have studied the process in several cell lines. In addition to HeLa cells, we have now included data from Jurkat cells in Fig1-2 and MAVER1 cells in the supplementary figure.

This is in particular important since HeLa cells do vary a lot between different laboratories and can have substantial cytogenetic defects. Along these lines it would be good to see another actin marker in addition to LifeAct (e.g. F-tractin) so substantiate the claim and to exclude possible artefacts emanating from the probe itself.

As mentioned, in addition to the Lifeact, we now provide cells (HeLa and Jurkat) immunostained for F-actin (with Phalloidin), tubulin and pericentrin (for centrosomes) at metaphase and anaphase.

Figure 2:

Throughout the manuscript the HeLa example given in Figure1 remains the only spontaneous physiological mitosis not involving long time drug manipulations such as performed the other figures. This STLC-arrested and Ro3306-mediated (Cdk1 inhibition) mitotic exit is a long term treatment. While this is surely a valid way to study some aspects of the cell cycle through drug treatments one would still like to see this phenomenon in a more physiological context.

As indicated above, we have changed Figure 1 to make the normal actin localisation during anaphase clearer. However, it is important that we carry out experiments that enable us to compare the actin-nucleation activity of isolated centrosomes in vivo and we have explained this better in the text.

If the authors would like to make the point that centrosomal actin assembly is critical for proper mitosis they should more convincingly show this in cells undergoing mitosis while adding Arp2/3 inhibitor under the microscope.

This is not possible. Many of the experiments can only be done reproducibly with cells that have been forced to exit mitosis synchronously – since treatments like the Arp2/3 inhibitor and WASH RNAi take time and inhibit mitotic entry and progression. Importantly, however, the accumulation of actin seen in cells forced to exit mitosis recapitulates that seen in control cells as shown in Figure 2.

In Fig 2B in DMSO control cells one can again only observe a cytoplasmic actin cloud. Efforts to improve imaging resolution may be desirable.

The zoom shows increase of actin around centrosomal region. This again we have confirmed with fixed images of cells stained with Phalloidin (F-actin) and tubulin. We have also done this analysis in Jurkat (Figure 2) and MAVER1 (Supp Fig1), which are now included in the revised manuscript.

Figure 3C, did the authors test whether other actin assembly factors colocalize with centrin1? Some specificity control would be good to see here. For example, GFP-WASP?

WASH is the relevant Arp2/3 activator. We show that WASH is required for all the actin accumulating at centrosomes and the cytoplasm in cells exiting mitosis. We have included the localization of WASH in MAVER1 cell line with cells arrested at prometaphase with STLC and also cells forced to exit mitosis following RO3306 treatment. It shows centrosome localization at these stages, further strengthening their role in this transient actin accumulation during mitotic exit.

Additionally, WASH has been previously shown to localise to centrosomes (unlike other actin nucleators).

Figure 4A, the proposed "shifting in size of strumpelin" (figure legend 4A) is not very convincing. There seems to be also no reference to strumpelin in the main text.

We have now focused on the shift in the size of the WASH1 protein, and have amended the text accordingly.

Did the authors look at Arp2/3 ko cells? Do they display this phenotype?

No. Arp2/3 activity and WASH is essential in most human cell lines including the ones used here. Moreover, given the importance of Arp2/3, KO lines are likely to have compensatory mutations.

Referee #2:

Farina et al. present some experiments which indicate that the actin nucleated at centrosomes at the end of mitosis downregulates microtubule nucleation. It's an interesting set of observations at the interface of these two cytoskeletal systems during cell division. The conclusions seem to me to be well supported by the data, taking advantage of in vitro and in cellulo approaches. I have some suggestions for the authors to improve the presentation of their paper (not presented in order of importance).

We thank the Reviewer for their assessment.

Correction for multiple comparisons: Experiments with more than two groups cannot be tested using multiple Student's t-tests, this is subject to increased likelihood of Type I error. An ANOVA with a suitable post-hoc test (e.g. Tukey's or Dunnet's with a single control) should be used instead. This applies if the data is normally distributed, please test for this first. This affects the statistical test in Fig 1C, Fig 3L; assuming what is written in the Methods is correct. The test should be declared in the legend. I'd encourage the authors to put the p-values on the graph rather than the star fest.

We appreciate the comment and have amended the manuscript accordingly.

In all graphs, markers and error bars should be defined in the legend (what do they show? mean/median {plus minus} SD/SEM/95%CI).

We have included the description in the figure legends.

Fig 1B As I understand it this plot shows data from the 2 cells shown in 1A, why are there error bars? N=1 for each line on the graph. I'm guessing it reflects multiple ROIs, multiple z-slices or the two centrosomes. If so it's not correct, bars should be removed.

We have placed Figure 1 with new panels.

Figs 2F,2H,3B,3D,4G suffer from too many points and oversized markers. They should make the markers smaller, use transparency or better still add box-whisker or violins to show the shape of the distribution. At the absolute minimum, they should shrink the markers to dots and include an error bar indicating the standard deviation, like the plot in 3L.

We have changed all the graphs.

Results 2nd para (page & line numbers would've been handy!): this reads a bit strangely. STLC and Cdk1 inhibitor are applied but there's no rationale given. "This changed when cells were forced to undergo..." this sounds strange - almost like the authors are surprised that someone has applied the drug. It would make more sense to say: The motivation is to better visualize centrosomal actin (the isolation part can be described later), so to do this we used STLC to blah blah and then forced exit by blah blah. The paper is generally easy to understand and well written, but this part could be improved. The FME method is used in all the subsequent figures so it feels like it should be better introduced.

We have amended the text and now clearly explain the rationale for the using forced mitotic exit.

Fig 4I legend should read anti-WASH1? The experiments are a depletion of WASH not supplementation.

Thanks. We have corrected this.

Referee #3:

This manuscript by Farina and colleague investigates the role of actin at centrosomes during mitotic exit. The authors suggest a model that a burst of Arp2/3-dependent actin nucleation at the centrosome at anaphase aids the disassembly of centrosomal microtubules (MTs). While this is an attractive model, the authors do not present sufficient evidence to support this model.

As stated above, in this revised manuscript, we have included better cell biology data in line with our *in vitro* work to support our claim.

This manuscript, similar to published data on *in vivo* centrosomal actin, suffers from two major problems 1) the low quality of the light microscopy, and 2) the inability to specifically perturb the actin nucleation machinery at centrosomes independent of the entire cell. The authors overinterpret their images, and their global drug treatments are used to strongly conclude specificity to centrosomes.

1) We have improved our cell biology data with high resolution images.
2) While we appreciate the difficulties of using global treatments to perturb actin in anaphase cells, it is clear from the *in vivo* work (although it was not very obvious in the images in the original Figure 1), that actin filaments arise at centrosomes in cells entering anaphase. In addition, the treatments block actin accumulation at centrosomes, both *in vitro* and *in vivo*, and in the cytoplasm. It is also likely that actin nucleated at centrosomes leads to more widespread actin accumulation via branch mediated Arp2/3 nucleation. To strengthen our argument, in our revised manuscript we have included data to show localisation of actin regulator WASH at centrosomes. We have also toned down the text to make it clear that formation of actin around centrosomes is, in some experiments, accompanied by more widespread actin filament formation.

I do not agree with their conclusions given the presented data.

We think the new data are much more convincing.

Major comments

Figure 1 shows live analysis and measurements of unperturbed cells. There are two concerns here:
1. The system: The authors attempt to measure actin and MTs at the centrosome, but the images presented are of the spindle pole region. This is due to the notorious HeLa cell rounding, which is why HeLa cells are generally avoided when studying MTs in mitosis. The authors should identify a more suitable system, or greatly improve their imaging. Imaging such as the following example of a HeLa cell is necessary to quantify the MT ring surrounding the centrosomes
<https://www.sciencedaily.com/releases/2010/03/100310091652.htm>

While we appreciate the importance of good imaging for observing actin at centrosomes, we have studied actin at centrosomes in both flat and round mitotic cells (something that we are expert in), and in a variety of cell lines. These studies have shown conclusively that i) HeLa and Jurkat cells are a good model for this analysis in which we can combine the analysis of actin nucleation *in vivo* and from isolated centrosomes and ii) that flat cells are not good for this type of analysis, as they increase the proximity of the cortex to the centrosomes *in z*.

We have replaced Figure 1 with stills from higher resolution movies and also provided data from fixed immunostainings of two different cell lines to support our claim.

2. The quantification. The authors should perform several control measurements of actin throughout the cell such as the cortex and different regions of the cytoplasm.

The focus of the paper however is on centrosomal actin. CK666 has little if any impact on cortical actin in cells exiting mitosis.

Additionally, the authors should treat unperturbed (no STLC) cells with CK666 and quantify changes in MTs and actin over time from their live movies.

CK666 prevents mitotic entry and progression, as does WASH RNAi. This is an additional reason for using monopolar exit as a model system for the analysis, in addition to the in vitro analysis.

Figure 2 There are two concerns here as well:

1. The system: To overcome the imaging limitation of HeLa cells, the authors use a two-drug system to isolate centrosomes away from the cortex (STLC) and drive cells out of mitosis (Ro-3306).

This is not correct. HeLa cells proved the best system to study actin nucleation at centrosomes for several reasons.

First, they are very round in mitosis, which helps. Second, they can be easily synchronised. Third, they are a great system for RNAi/molecular genetics.

However, in our revised manuscript we have included the data showing actin at anaphase centrosomes in Jurkat and MAVER cell lines.

Thus, the controls used in this study are cancer cells that are not amenable to imaging and have been treated with two drugs. Although, the images in 2A are the most convincing evidence of an in vivo centrosome having the capability of nucleating actin, these cells are heavily manipulated, throwing into question the relevance of this observation.

We disagree with this. There are thousands of studies of mitosis in HeLa cells.

Moreover, there is no such thing as a normal human cell in culture. All cell lines are model systems. In addition, we now include data from Jurkat and MAVER cell lines which show the same trend.

2. The quantification: The authors should present both Actin and MT measurements for these live experiments. It would also be helpful if the intensity axes are labeled with what is measured.

We have included this data.

The most telling statement in the manuscript, and one that concedes that no specificity can be determined using these global drug treatment is "Just as observed in cells treated with CK666, the silencing of WASH by RNAi led to the near complete loss of cytoplasmic actin in cells exiting mitosis, as assayed using both live imaging (Figure 4C-D) and in cells that had been fixed and stained for F-actin (Figure 4F-G)."

We have aimed to describe the result as observed.

Actin is seen at centrosomes in vivo and in vitro in anaphase. This is blocked by CK666 and WASH RNAi, along with cytoplasmic actin. Much of this may be generated via Arp2/3 side-branching activity. However, the focus of our analysis is on the crosstalk between actin and microtubules at centrosomes, making the centrosomal pool relevant and important. To make this clearer, we have now extended the discussion in the paper describing more widespread cytoplasmic actin filament formation.

Other comments

a. In the introduction, the authors refer to MT dynamics in mitosis and reference Mchedlishvili. This manuscript does not quantify MT dynamics, which is a specific term used to describe events of growing, shrinking, pausing, catastrophe and rescue. Better references to changes in MT dynamics are found from the Wadsworth lab from the early 2000s. Similarly, at the end of the introduction the authors state they have identified a role for actin nucleation in the regulation of mitotic microtubule dynamics. This is not an accurate representation of the work.

This was a mistake. We meant to refer to dynamic changes in microtubule organisation, for which the reference is appropriate. This has been changed.

b. The following statement is an unsubstantiated conclusion from the previous paper, which was also unable to show specific loss of Arp2.3 at the centrosome - "Since previous work demonstrated that the actin formed at interphase centrosomes was nucleated by a pool of Arp2.3 present at centrosomes (Farina et al., 2016)."

We will remove the final "at centrosomes". However, we would like to make it clear that i) Arp2.3 is localised at centrosomes and ii) is required for actin nucleation at centrosomes.

c. The following two relevant statements are conjecture - "In metaphase cells, this revealed a low level of cytoplasmic actin". The images from Figure 1 show substantial cytoplasmic actin.

We will show better images to support this.

"As previously noted, CK666 did not have a profound impact on the formation of the mitotic cortical actin mesh." Is there a specific reason to suspect that only profound changes in actin would result in meaningful changes of cellular physiology? Cortical F-Actin should be measured in untreated and CK666 treated cells.

We have removed this statement as it is irrelevant to our analysis, which is focused on non-cortical actin.

2nd Editorial Decision

10th January 2019

I am sorry for the delay in getting back to you with a decision on your revised manuscript. The re-reviews were not straightforward, and we therefore needed time to further discuss them in our team, including our Chief Editor Bernd Pulverer - all this being slow over the holiday period and the peak submission time in December. As it turned out, neither referees 1 nor 3 were really satisfied with your responses and revisions, leaving only the (already originally positive) reviewer 2 overall supportive - although even s/he agreed during referee cross-commenting that the revisions could have been more exhaustive. Reviewer 1 still has a number of issues with the data as well as conceptual concerns; while reviewer 3 (whose detailed report is attached to this letter in PDF format) is generally not convinced that the new version offers now stronger evidence for centrosomal actin influence on microtubules during mitosis.

In light of these still critical opinions from a majority of reviewers even after a considerable revision period, we find it difficult to envision the study easily becoming a more promising EMBO Journal candidate without extensive further revision rounds. Given that we usually do not consider multiple major revisions reasonable and in the best interest of authors, I am afraid we therefore had to conclude that we will not be able to publish this work in The EMBO Journal. However, I took the liberty to discuss the manuscripts and its reviews with my EMBO reports colleague, who expressed strong interest in publishing a re-revised version with only minor final modifications. Should you be interested in this option, please simply use the link below to directly transfer the study including referee reports to EMBO reports. Obviously, Deniz would also be happy to further discuss the process and the revision/amendments needed with you at any point - she can be contacted at d.senyilmaz@emboreports.org

I am sorry that I could not be more positive for The EMBO Journal on this occasion, especially after your revision efforts and the prolonged re-evaluation, but would nevertheless like to thank you again for having had the opportunity to consider this work, and very much hope you find the possibility of rapid publication in EMBO reports worthwhile.

REFEREE REPORTS.

Referee #1:

This is a revised manuscript.

The major point remains that there is no mechanism for how "local actin nucleation (is) to tune the levels of centrosomal microtubules " and that the imaging quality fails to convince me.

The authors state in their rebuttal that studying spontaneous mitosis is not possible. I still disagree and don't understand the reasoning for not investigating spontaneous mitosis while performing drug additions on living cells under the microscope.

"How this mutual antagonism between local actin and microtubule formation at the centrosome works remains to be determined ". This remains problematic and as such the papers remains descriptive in large parts. siRNA treatments have also not been rescued.

The authors further write:

".as the two systems compete for physical space around the centrosome."

This is entirely speculative. How can they exclude a cooperative function instead?

Specific points:

Figure 1C, I simply cannot see any centrosomal actin. Even in 1E I have a hard time to follow.

Figure 2A is lacking tubulin alone. To me it seems that there is substantial microtubule outgrowth, eve though only a short time window is depicted. Hence, I cannot follow the local inhibition hypothesis.

Figure 3A. left control panel. I cannot see any centrosomal actin. For me this looks simply cytoplasmic.

Figure 5 should corroborated using a kinase inhibitor. As now it is merely an observation.

Figure 6A. control control panel. I cannot see any centrosomal actin. For me this looks simply cytoplasmic. Resolution is simply not good enough to convince this reviewer.

Figure 6. siWASH should be rescued. This should be standard procedure in siRNA experiments.

Referee #2:

In my original report, I felt that the manuscript was an interesting set of observations that would be of interest to the readership of EMBO J and still have that opinion. I could see that the other referees were more critical, but I was pleased that you invited a resubmission. As far as my comments are concerned, the authors have addressed the points that I raised. I think that they have made some effort to address the critiques of the other reviewers. Particularly showing that the phenomena are not limited to a single cell line. I have no new comments to add.

Referee #3:

Review of resubmitted manuscript

I have now carefully read the revised manuscript by Farina et al and the response to reviewers.

Unfortunately, my opinion has not changed as each of my concerns from the previous version have not been addressed. The author failed to present convincing data that actin influences microtubules at centrosome. At best, the authors show a correlation in wild type cell.

I have submitted my complete review via email as the response to each rebuttal point will not be clear by pasting into this text box.

Review of resubmitted manuscript

I have now carefully read the revised manuscript by Farina et al and the response to reviewers.

Unfortunately, my opinion has not changed as each of my concerns from the previous version have

not been addressed. The author failed to present convincing data that actin influences microtubules at centrosome. At best, the authors show a correlation in wild type cell.

Point by point response (new comments in black)

Referee #3: This manuscript by Farina and colleague investigates the role of actin at centrosomes during mitotic exit. The authors suggest a model that a burst of Arp2/3-dependent actin nucleation at the centrosome at anaphase aids the disassembly of centrosomal microtubules (MTs). While this is an attractive model, the authors do not present sufficient evidence to support this model.

As stated above, in this revised manuscript, we have included better cell biology data in line with our in vitro work to support our claim.

This manuscript, similar to published data on in vivo centrosomal actin, suffers from two major problems 1) the low quality of the light microscopy, and 2) the inability to specifically perturb the actin nucleation machinery at centrosomes independent of the entire cell. The authors over interpret their images, and their global drug treatments are used to strongly conclude specificity to centrosomes.

1) We have improved our cell biology data with high resolution images.

2) While we appreciate the difficulties of using global treatments to perturb actin in anaphase cells, it is clear from the in vivo work (although it was not very obvious in the images in the original Figure 1), that actin filaments arise at centrosomes in cells entering anaphase. In addition, the treatments block actin accumulation at centrosomes, both in vitro and in vivo, and in the cytoplasm. It is also likely that actin nucleated at centrosomes leads to more widespread actin accumulation via branch mediated Arp2/3 nucleation. To strengthen our argument, in our revised manuscript we have included data to show localisation of actin regulator WASH at centrosomes. We have also toned down the text to make it clear that formation of actin around centrosomes is, in some experiments, accompanied by more widespread actin filament formation.

I do not agree with their conclusions given the presented data.

We think the new data are much more convincing.

Major comments

Figure 1 shows live analysis and measurements of unperturbed cells. There are two concerns here:

1. The system: The authors attempt to measure actin and MTs at the centrosome, but the images presented are of the spindle pole region. This is due to the notorious HeLa cell rounding, which is why HeLa cells are generally avoided when studying MTs in mitosis. The authors should identify a more suitable system, or greatly improve their imaging. Imaging such as the following example of a HeLa cell is necessary

*to quantify the MT ring surrounding the centrosomes
<https://www.sciencedaily.com/releases/2010/03/100310091652.htm>*

While we appreciate the importance of good imaging for observing actin at centrosomes, we have studied actin at centrosomes in both flat and round mitotic cells (something that we are expert in), and in a variety of cell lines. These studies have shown conclusively that i) HeLa and Jurkat cells are a good model for this analysis in which we can combine the analysis of actin nucleation in vivo and from isolated centrosomes and ii) that flat cells are not good for this type of analysis, as they increase the proximity of the cortex to the centrosomes in z. We have replaced Figure 1 with stills from higher resolution movies and also provided data from fixed immunostainings of two different cell lines to support our claim.

While the authors have studied HeLa cells in the past and are experts, it does not make HeLa cells good for imaging. The argument against using a flat cells is not convincing as the authors do not state why being closer to the cortex might change actin behavior. Furthermore, rounded HeLa cells bring the lateral cortex in very close proximity to the centrosome (as you can see in their images), probably within 1-2 microns which is similar to what happens in a flat cells.

2. The quantification. The authors should perform several control measurements of actin throughout the cell such as the cortex and different regions of the cytoplasm.

The focus of the paper however is on centrosomal actin. CK666 has little if any impact on cortical actin in cells exiting mitosis.

This is not an acceptable response. The authors are arguing for changes in centrosomal actin as a specific mechanism to control microtubule levels at the centrosome. Showing that f-actin levels globally do not change is very important. While measurements at the cortex might not be critical, measurement within different regions of the cytoplasm is very important. If actin levels change in other regions of the cell as one can see in figure 1A several microns away from the centrosome, then the arguments for a specific mechanism at the centrosome becomes less compelling. Also, my comment is not only related to the CK666 treatment, but to each figure in the manuscript. Figure 1E for example shows very nicely that the cytoplasmic levels of actin are indistinguishable from that of the centrosomes, with the exception one frame at minute 5 of C2, which appears to be a random blob actin that could likely form anywhere, exactly as shown in Figure 1C away from the centrosome.

Additionally, the authors should treat unperturbed (no STLC) cells with CK666 and quantify changes in MTs and actin over time from their live movies.

CK666 prevents mitotic entry and progression, as does WASH RNAi. This is an additional reason for using monopolar exit as a model system for the analysis, in addition to the in vitro analysis.

Again here the authors fail to perform controls and instead argue that they are not needed. The authors should treat cells with each inhibitor, find cells that are already in mitosis and report what happens to actin around the centrosome and in the cytoplasm. Likewise, treatment with CK666 or WASH RNAi, should be followed by Cdk1 inhibition and then actin should be measured. The main point here is that drug treatments need additional controls.

Figure 2 There are two concerns here as well: 1. The system: To overcome the imaging limitation of HeLa cells, the authors use a two-drug system to isolate centrosomes away from the cortex (STLC) and drive cells out of mitosis (Ro-3306).

This is not correct. HeLa cells proved the best system to study actin nucleation at centrosomes for several reasons. First, they are very round in mitosis, which helps. Second, they can be easily synchronised. Third, they are a great system for RNAi/molecular genetics. However, in our revised manuscript we have included the data showing actin at anaphase centrosomes in Jurkat and MAVER cell lines.

Each of the three reasons presented is not an advantage. The authors state that the fact they are round helps, but they don't say why. If it is in reference to the distance of the centrosome from the cortex, then I've addressed this above. The second two reasons can be done in many other cell lines.

Thus, the controls used in this study are cancer cells that are not amenable to imaging and have been treated with two drugs. Although, the images in 2A are the most convincing evidence of an in vivo centrosome having the capability of nucleating actin, these cells are heavily manipulated, throwing into question the relevance of this observation.

We disagree with this. There are thousands of studies of mitosis in HeLa cells. Moreover, there is no such thing as a normal human cell in culture. All cell lines are model systems. In addition, we now include data from Jurkat and MAVER cell lines which show the same trend.

There is really nothing with which to disagree. The thousands of studies that use HeLa cells equally suffer from the same imaging issues and genome instability issues as this study. Several studies have now shown that each lab has their own version of HeLa cells. While no cell line is normal, there are several that are highly regarded as non-cancer cells. But, we digress.

My comment here was more focused on the control situation being a cell treated with two drugs that have global effects.

2. The quantification: The authors should present both Actin and MT measurements for these live experiments. It would also be helpful if the intensity axes are labeled with what is measured.

We have included this data.

The authors do not present quantification for microtubules for figures 2, 3, and 6. The main point of this study is to attempt to link actin levels and regulation at the centrosome with centrosomal microtubules. Thus, microtubule measurements should be presented.

The most telling statement in the manuscript, and one that concedes that no specificity can be determined using these global drug treatment is "Just as observed in cells treated with CK666, the silencing of WASH by RNAi led to the near complete loss of cytoplasmic actin in cells exiting mitosis, as assayed using both live imaging (Figure 4C-D) and in cells that had been fixed and stained for F-actin (Figure 4F-G)."

We have aimed to describe the result as observed. Actin is seen at centrosomes in vivo and in vitro in anaphase. This is blocked by CK666 and WASH RNAi, along with cytoplasmic actin. Much of this may be

generated via Arp2/3 sidebranching activity. However, the focus of our analysis is on the crosstalk between actin and microtubules at centrosomes, making the centrosomal pool relevant and important. To make this clearer, we have now extended the discussion in the paper describing more widespread cytoplasmic actin filament formation.

Other comments

a. In the introduction, the authors refer to MT dynamics in mitosis and reference Mchedlishvili. This manuscript does not quantify MT dynamics, which is a specific term used to describe events of growing, shrinking, pausing, catastrophe and rescue. Better references to changes in MT dynamics are found from the Wadsworth lab from the early 2000s. Similarly, at the end of the introduction the authors state they have identified a role for actin nucleation in the regulation of mitotic microtubule dynamics. This is not an accurate representation of the work.

This was a mistake. We meant to refer to dynamic changes in microtubule organisation, for which the reference is appropriate. This has been changed.

b. The following statement is an unsubstantiated conclusion from the previous paper, which was also unable to show specific loss of Arp2.3 at the centrosome - "Since previous work demonstrated that the actin formed at interphase centrosomes was nucleated by a pool of Arp2.3 present at centrosomes (Farina et al., 2016)."

We will remove the final "at centrosomes". However, we would like to make it clear that i) Arp2.3 is localised at centrosomes and ii) is required for actin nucleation at centrosomes.

c. The following two relevant statements are conjecture - "In metaphase cells, this revealed a low level of cytoplasmic actin". The images from Figure 1 show substantial cytoplasmic actin.

We will show better images to support this.

Measurements are still needed for cytoplasmic actin at various distances from the centrosome

"As previously noted, CK666 did not have a profound impact on the formation of the mitotic cortical actin mesh." Is there a specific reason to suspect that only profound changes in actin would result in meaningful changes of cellular physiology? Cortical F-Actin should be measured in untreated and CK666 treated cells.

We have removed this statement as it is irrelevant to our analysis, which is focused on noncortical actin.

Author Correspondence

16th January 2018

Thank you again for agreeing to review our paper in EMBO Journal and for getting back to us after the latest round of reviews.

Having read the reviews carefully, we think that they are very poor.

This is disappointing and is not the kind of critical review we expected from a journal of the caliber of EMBO.

Reviewers 1 and 3 have not made specific comments that we could address to improve the paper. Instead, in response to our revision, they seem to want us to re-explain what would be obvious to an expert. In addition, they say they are "not convinced" by the data we provide. This is not good enough in a review. The data speaks for itself. Our results are clear and reproducible across systems.

In this regard, a major problem seems to be a desire of the Reviewers for us to show that actin is ONLY nucleated around centrosomes in anaphase. This is odd, since this is not what we see, isn't what we say we see, and isn't what we propose happens at anaphase. In the paper, we acknowledge the presence of cytoplasmic actin and suggest that the actin nucleation at the centrosomes could lead to widespread actin accumulation in the cytoplasm. However, our in vitro work clearly suggests that centrosomes are a likely source of this cytoplasmic pool of actin that is centered about centrosomes at anaphase. This has been quantified again and again for different experiments presented in the paper. Thus, our conclusions are completely in line with the quantitative data. If necessary to include, we have data that show that the increase in actin at anaphase is higher at centrosomes than at control regions of the cytoplasm.

We can only conclude that these two Reviewers are prejudiced in their reading of the paper or have not understood the data. In line with the former, Reviewer 3 appears not to believe previous published data on centrosomal actin in interphase. Again, this is a problem. It's not a matter of belief, but a matter of data.

All the results are consistent in showing that there is a transient burst of WASH and Arp2.3 dependent actin accumulation that is nucleated close to, (but is not confined to) centrosomes at anaphase, which plays a role in modifying centrosomal microtubule nucleation. We have now replicated these findings in multiple cell types, using live imaging, fixed imaging, in cycling cells and under conditions of forced mitotic exit, and in biochemical experiments. Moreover, perturbations have been carried out using RNAi, small molecule inhibitors and blocking antibodies. We think this constitutes a rigorous analysis. Moreover, for the revision, we repeated all the data presented in Figure 1. The quantitative results of this repeat are completely in line with the data presented in the first draft.

Additionally, in our last submission we clearly explained and justified our use of monopolar cytokinesis (or forced mitotic exit). Our data shows that the actin accumulation around centrosomes is transient, therefore it is important to have cells at the same phase of mitotic exit for proper

quantification. Drug treatments take time to act and alter the timing of mitotic exit. Forced mitotic exit allows us to synchronize cells and perform quantifications under these conditions. We have also clearly shown that the dynamics of actin accumulation during forced mitotic exit are similar to regular mitosis in different cell lines (Hela, Jurkat, MAVER1).

Reviewer 3 also has some specific questions about the cells we are using for the experiments. Reviewer 3 does not believe that Hela cells are a good model system for studying actin cytoskeleton during mitosis as they are very round. This is frankly ridiculous. They should read the literature! The reviewer's view that Hela cells are not an ideal model system cannot be addressed rationally. As we have explained multiple times, when cells are flat, the centrosomes come in close proximity to the cortex (in Z), making it difficult to distinguish between centrosomal pool and cortical actin network (which is very bright). Thus, for our analysis, we ensured that when measured, the centrosomes are further away from the cortex, to avoid errors in the measurement.

Finally, Reviewer 1 is not satisfied with our statement "How this mutual antagonism between local actin and microtubule formation at the centrosome works remains to be determined ". We think it is clear that identifying the biophysical mechanisms that underlie the antagonism between the two systems is well beyond the scope of this paper. It would take 4 additional years.

While we can add a few more minor pieces of supporting data (e.g. data to show that the increase in actin at anaphase is higher at centrosomes than at control regions of the cytoplasm), but we do not think there is more that can be done to improve the study to convince Reviewers 1 and 3.

In light of this, perhaps EMBO J should invite an additional reviewer to look at the whole process before suggesting EMBO Reports as a fall-back position.

Editor Correspondence

24th January 2018

I have now looked at your response, and to some degree understand your frustration with the reports. I also agree that it cannot be expected to extend the study into multiple further directions such as deeper mechanisms or repetitions in other cell lines. That leaves the key question whether the data as presented support the key conclusions of the paper, and here there seems to be a bit of a deadlock, with two referees not being convinced about that (on two counts!) and you as an author obviously disagreeing with that opinion.

I discussed all this once more with our Chief Editor and we felt that in this situation, the most appropriate thing to do would indeed be involving an additional, unbiased arbitrator with expertise in mitotic spindle function - I would provide them the manuscript in its latest version, together with the first-round comments and your response to them, and ask whether they would consider the paper compelling and the original concerns adequately addressed; I may also hint at the remaining concrete criticisms (data conclusiveness, cell system etc.) to get the advisor's view on these contested issues.

While I obviously cannot predict/guarantee the outcome of this arbitration, I hope you will find this an acceptable way forward at this stage; the EMBO reports transfer offer remaining unaffected by this for now. Also, please understand that it may take some days for me to get a most trusted advisor of our journal assigned to this task.

3rd Editorial Decision

14th March 2019

Thank you for your patience while we have been soliciting additional expert opinions on your submission, and further discussing this feedback internally.

A fourth, arbitrating referee (whose detailed comments are copied below) has now looked at your revised manuscript, as well as at the initial referee reports from the first round and your responses to them. As you will, referee 4 feels that you have overall answered the earlier concerns of all three original reviewers in a satisfactory manner. At the same time, the arbitrator does share the

reservations of referees 1 and 3 regarding the still somewhat unclear functional relevance of centrosomal actin nucleation during mitosis, albeit admitting that this may not be straightforward to elucidate.

Given that we initially invited revision of the manuscript, and that the arbitrating referee's assessment of your responses to the original points indicates that the continued specific reservations of referees 1 and 3 may not warrant dismissing the study at this stage, we eventually decided to accept the study for publication pending adequate modifications in a final (minor) revision round. Importantly, in these ultimate revisions:

- please sensitively comment on each point raised by referees 1 and 3 (including those in their additional PDF report) in the previous round, as well as on the conceptual caveats pointed out by the new referee 4. This should be done both in a point-by-point response and with according changes to the text/figures as appropriate. Given the shared concerns from previous and new reviewers, we would find it important to tone down the conclusions on physiological function to make sure that this is not overstated and any limitations are clearly noted.

- where relevant, please include additional control/supporting data (such as the data to show that the increase in actin at anaphase is higher at centrosomes than at control regions of the cytoplasm, which you mentioned in an earlier message).

- please consider uploading some of the provided movies as "Expanded View Movies" and using them in the text as supporting evidence. Movies should be uploaded individually as ZIP files, each containing one movie file and its respective legend text file, and referenced in the text as "Movie EV1" etc

REFeree REPORTS.

Referee #4:

I think that the authors have addressed satisfactorily the concerns of reviewer 1 and 2 and this has definitely improved very much the manuscript providing more convincing data and a better quality of analysis.

Concerning reviewer 3, I found that several of the criticisms were not really scientifically sound and I support the authors when they disagree or state they are not correct. However the authors did also reply correctly to other specific comments and this also has improved their manuscript.

Overall I think that the results are supported by the data presented. However what is not entirely clear to me is how relevant this increase in actin nucleation at the centrosome (still relatively mild although apparently specific to anaphase) is. There are no experimental data that really address this issue. I am of course fully aware that this would not be trivial or maybe even doable in a convincing way at the moment. But this leaves the whole story as a quite descriptive work and speculations on possible functional implications. The proposal from the authors is that the increase of actin nucleation at the centrosome in anaphase may be important to lower microtubule nucleation and help in the establishment of the telophase microtubule network. It may be the case but this is really quite a speculation at this point.

In summary I think that the authors answered correctly to the reviewers, improving their manuscript. Still the main concerns of reviewer 1 and 3 is about the relevance of the data, reviewer 1 phrases it as relevance for 'physiological mitosis', reviewer 3 is not convinced by the conclusion that actin nucleation aids the disassembly of centrosomal microtubules. These issues are both pointing at the lack of functional data and were not addressed by the authors.

Reviewer1 comments-

This is a revised manuscript.

The major point remains that there is no mechanism for how "local actin nucleation (is) to tune the levels of centrosomal microtubules " and that the imaging quality fails to convince me.

The authors state in their rebuttal that studying spontaneous mitosis is not possible. I still disagree and don't understand the reasoning for not investigating spontaneous mitosis while performing drug additions on living cells under the microscope.

In the revised manuscript we have included data on mitosis from different cell lines and clearly explain our use of forced mitotic exit. In addition, as per the reviewer's suggestion, we have included data on Arp2.3 inhibition during bipolar mitosis, which shows a similar trend to the monopolar forced exit described earlier.

"How this mutual antagonism between local actin and microtubule formation at the centrosome works remains to be determined ". This remains problematic and as such the papers remains descriptive in large parts. siRNA treatments have also not been rescued.

The authors further write:

"..as the two systems compete for physical space around the centrosome."

This is entirely speculative. How can they exclude a cooperative function instead?

We have now provided data for tubulin levels during forced mitotic exit, with/without Arp2/3 inhibition (monopolar and bipolar) to support our hypothesis of mutual antagonism. Further, we have expanded our discussion of possible mechanisms in the manuscript.

In addition to using WASH siRNA treatment, we have performed experiments with

1. WASH antibody (*in vitro*)
2. Arp2/3 inhibitor (*in vivo* and *in vitro*)

We have shown using independent methods both *in vitro* and *in vivo* that these proteins are involved in the transient actin accumulation.

Specific points:

Figure 1C, I simply cannot see any centrosomal actin. Even in 1E I have a hard time to follow.

We have now included a higher resolution image to show actin accumulation around the centrosomal region. Further, we have included a panel of cells labelled with different probes – siR-actin, Lifeact-GFP and Lifeact-mCherry, which demonstrate this transient accumulation of actin around the centrosomes.

The reviewer is correct in pointing out that there seems to be an increase in cytoplasmic actin during anaphase. We have included measurements of non-centrosomal (cytoplasmic actin) in HeLa cells to determine the regional specificity of actin accumulation. We find that while there is some increase in cytoplasmic actin during anaphase, it is not significantly different from the pool at metaphase. Further, the increase around centrosomal region is significantly higher than the cytoplasmic pool at anaphase (both in live and fixed HeLa cells). We have added this data to further corroborate the specificity of actin accumulation during anaphase.

Figure 2A is lacking tubulin alone. To me it seems that there is substantial microtubule outgrowth, even though only a short time window is depicted. Hence, I cannot follow the local inhibition hypothesis.

As mentioned above, we have provided data for tubulin levels during forced mitotic exit, with/without Arp2/3 inhibition (monopolar and bipolar) to support our hypothesis of mutual antagonism. We acknowledge that we are concluding the premise of our model from the amounts of

actin and tubulin rather than their rate of polymerization. Therefore, we have modified our discussion to reflect this.

Figure 3A. left control panel. I cannot see any centrosomal actin. For me this looks simply cytoplasmic.

The non-centrosomal cytoplasmic actin quantification should make the difference clearer. Further, we have included a panel in the supplementary figures with timelapse images from multiple cells showing actin accumulation specifically around the centrosomal region during forced mitotic exit.

Figure 5 should corroborated using a kinase inhibitor. As now it is merely an observation.

Our data shows that the WASH complex protein undergoes changes in phosphorylation state and size during mitosis. The panel (B) shows that on addition of Cdk1 inhibitor to induce forced mitotic exit, these changes seem to be reversed.

Figure 6A. control control panel. I cannot see any centrosomal actin. For me this looks simply cytoplasmic. Resolution is simply not good enough to convince this reviewer.

Figure 6. siWASH should be rescued. This should be standard procedure in siRNA experiments.

As mentioned before, we have used different methods to confirm the WASH siRNA phenotype.

Reviewer 3 comments-

I have now carefully read the revised manuscript by Farina et al and the response to reviewers. Unfortunately, my opinion has not changed as each of my concerns from the previous version have not been addressed. The author failed to present convincing data that actin influences microtubules at centrosome. At best, the authors show a correlation in wild type cell.

Point by point response (new comments in black)

Referee #3: This manuscript by Farina and colleague investigates the role of actin at centrosomes during mitotic exit. The authors suggest a model that a burst of Arp2/3-dependent actin nucleation at the centrosome at anaphase aids the disassembly of centrosomal microtubules (MTs). While this is an attractive model, the authors do not present sufficient evidence to support this model.

As stated above, in this revised manuscript, we have included better cell biology data in line with our *in vitro* work to support our claim.

This manuscript, similar to published data on *in vivo* centrosomal actin, suffers from two major problems

1) the low quality of the light microscopy, and 2) the inability to specifically perturb the actin nucleation machinery at centrosomes independent of the entire cell. The authors overinterpret their images, and their global drug treatments are used to strongly conclude specificity to centrosomes.

1) We have improved our cell biology data with high resolution images.

2) While we appreciate the difficulties of using global treatments to perturb actin in anaphase cells, it is clear from the *in vivo* work (although it was not very obvious in the images in the original Figure 1), that actin filaments arise at centrosomes in cells entering anaphase. In addition, the treatments block actin accumulation at centrosomes, both *in vitro* and *in vivo*, and in the cytoplasm. It is also likely that actin nucleated at centrosomes leads to more widespread actin accumulation via branch mediated Arp2/3 nucleation. To strengthen our argument, in our revised manuscript we have included data to show localisation of actin regulator WASH at centrosomes. We have also toned down the text to make it clear that formation of actin around centrosomes is, in some experiments, accompanied by more widespread actin filament formation.

I do not agree with their conclusions given the presented data.

We think the new data are much more convincing.

Major comments

Figure 1 shows live analysis and measurements of unperturbed cells. There are two concerns here:
1. The system: The authors attempt to measure actin and MTs at the centrosome, but the images presented are of the spindle pole region. This is due to the notorious HeLa cell rounding, which is why HeLa cells are generally avoided when studying MTs in mitosis. The authors should identify a more suitable system, or greatly improve their imaging. Imaging such as the following example of a HeLa cell is necessary to quantify the MT ring surrounding the centrosomes
<https://www.sciencedaily.com/releases/2010/03/100310091652.htm>

While we appreciate the importance of good imaging for observing actin at centrosomes, we have studied actin at centrosomes in both flat and round mitotic cells (something that we are expert in), and in a variety of cell lines. These studies have shown conclusively that i) HeLa and Jurkat cells are a good model for this analysis in which we can combine the analysis of actin nucleation in vivo and from isolated centrosomes and ii) that flat cells are not good for this type of analysis, as they increase the proximity of the cortex to the centrosomes in z. We have replaced Figure 1 with stills from higher resolution movies and also provided data from fixed immunostainings of two different cell lines to support our claim.

While the authors have studied HeLa cells in the past and are experts, it does not make HeLa cells good for imaging. The argument against using a flat cells is not convincing as the authors do not state why being closer to the cortex might change actin behavior. Furthermore, rounded HeLa cells bring the lateral cortex in very close proximity to the centrosome (as you can see in their images), probably within 1-2 micros which is similar to what happens in a flat cells.

In light of the reviewer's comments, we performed experiments in other cell lines namely Jurkat and MAVER1 and find the transient accumulation of actin around centrosomes during anaphase.

2. The quantification. The authors should perform several control measurements of actin throughout the cell such as the cortex and different regions of the cytoplasm.

The focus of the paper however is on centrosomal actin. CK666 has little if any impact on cortical actin in cells exiting mitosis.

This is not an acceptable response. The authors are arguing for changes in centrosomal actin as a specific mechanism to control microtubule levels at the centrosome. Showing that f-actin levels globally do not change is very important. While measurements at the cortex might not be critical, measurement within different regions of the cytoplasm is very important. If actin levels change in other regions of the cell as one can see in figure 1A several microns away from the centrosome, then the arguments for a specific mechanism at the centrosome becomes less compelling. Also, my comment is not only related to the CK666 treatment, but to each figure in the manuscript. Figure 1E for example shows very nicely that the cytoplasmic levels of actin are indistinguishable from that of the centrosomes, with the exception one frame at minute 5 of C2, which appears to be a random blob actin that could likely form anywhere, exactly as shown in Figure 1C away from the centrosome.

We have measured non-centrosomal cytoplasmic actin and found no significant difference between metaphase and early anaphase, both in live and fixed HeLa cells.

We will add this data to further corroborate the specificity of actin accumulation around centrosomes during anaphase.

Additionally, the authors should treat unperturbed (no STLC) cells with CK666 and quantify changes in MTs and actin over time from their live movies.

CK666 prevents mitotic entry and progression, as does WASH RNAi. This is an additional reason for using monopolar exit as a model system for the analysis, in addition to the in vitro analysis.

Again here the authors fail to perform controls and instead argue that they are not needed. The authors should treat cells with each inhibitor, find cells that are already in mitosis and report what happens to actin around the centrosome and in the cytoplasm. Likewise, treatment with CK666 or WASH RNAi, should be followed by Cdk1 inhibition and then actin should be measured. The main point here is that drug treatments need additional controls.

We thank them for their suggestion. We have now included data on Arp2.3 inhibition during bipolar mitosis, which shows a similar trend to the monopolar forced exit described earlier.

Figure 2 There are two concerns here as well: 1. The system: To overcome the imaging limitation of HeLa cells, the authors use a two-drug system to isolate centrosomes away from the cortex (STLC) and drive cells out of mitosis (Ro-3306).

This is not correct. HeLa cells proved the best system to study actin nucleation at centrosomes for several reasons. First, they are very round in mitosis, which helps. Second, they can be easily synchronised. Third, they are a great system for RNAi/molecular genetics. However, in our revised manuscript we have included the data showing actin at anaphase centrosomes in Jurkat and MAVER cell lines.

Each of the three reasons presented is not an advantage. The authors state that the fact they are round helps, but they don't say why. If it is in reference to the distance of the centrosome from the cortex,

then I've addressed this above. The second two reasons can be done in many other cells lines.

We have done measurements in Jurkat and MAVER1 cell lines and observed similar increase in actin around centrosomes during anaphase

Thus, the controls used in this study are cancer cells that are not amenable to imaging and have been treated with two drugs. Although, the images in 2A are the most convincing evidence of an in vivo centrosome having the capability of nucleating actin, these cells are heavily manipulated, throwing into question the relevance of this observation.

We disagree with this. There are thousands of studies of mitosis in HeLa cells. Moreover, there is no such thing as a normal human cell in culture. All cell lines are model systems. In addition, we now include data from Jurkat and MAVER cell lines which show the same trend.

There is really nothing with which to disagree. The thousands of studies that use HeLa cells equally suffer from the same imaging issues and genome instability issues as this study. Several studies have now shown that each lab has their own version of HeLa cells. While no cell line is normal, there are several that are highly regarded as non-cancer cells. But, we digress. My comment here was more focused on the control situation being a cell treated with two drugs that have global effects.

2. The quantification: The authors should present both Actin and MT measurements for these live experiments. It would also be helpful if the intensity axes are labeled with what is measured.

We have included this data.

The authors do not present quantification for microtubules for figures 2, 3, and 6. The main point of this study is to attempt to link actin levels and regulation at the centrosome with centrosomal microtubules. Thus, microtubule measurements should be presented.

We have now provided data for tubulin levels during forced mitotic exit, with/without Arp2/3 inhibition (monopolar and bipolar) to support our hypothesis of mutual antagonism. Figure 3, EV5-6

The most telling statement in the manuscript, and one that concedes that no specificity can be determined using these global drug treatment is "Just as observed in cells treated with CK666, the silencing of WASH by RNAi led to the near complete loss of cytoplasmic actin in cells exiting mitosis, as assayed using both live imaging (Figure 4C-D) and in cells that had been fixed and stained for F-actin (Figure 4F-G)."

We have aimed to describe the result as observed. Actin is seen at centrosomes in vivo and in vitro in anaphase. This is blocked by CK666 and WASH RNAi, along with cytoplasmic actin. Much of this may be generated via Arp2/3 sidebranching activity. However, the focus of our analysis is on the crosstalk between actin and microtubules at centrosomes, making the centrosomal pool relevant and important. To make this clearer, we have now extended the discussion in the paper describing more widespread cytoplasmic actin filament formation.

Other comments

a. In the introduction, the authors refer to MT dynamics in mitosis and reference Mchedlishvili. This manuscript does not quantify MT dynamics, which is a specific term used to describe events of growing, shrinking, pausing, catastrophe and rescue. Better references to changes in MT dynamics are found from the Wadsworth lab from the early 2000s. Similarly, at the end of the introduction the authors state they have identified a role for actin nucleation in the regulation of mitotic microtubule dynamics. This is not an accurate representation of the work.

This was a mistake. We meant to refer to dynamic changes in microtubule organisation, for which the reference is appropriate. This has been changed.

b. The following statement is an unsubstantiated conclusion from the previous paper, which was also unable to show specific loss of Arp2.3 at the centrosome - "Since previous work demonstrated that the actin formed at interphase centrosomes was nucleated by a pool of Arp2.3 present at centrosomes (Farina et al., 2016)."

We will remove the final "at centrosomes". However, we would like to make it clear that i) Arp2.3 is

localised at centrosomes and ii) is required for actin nucleation at centrosomes.

c. The following two relevant statements are conjecture - "In metaphase cells, this revealed a low level of cytoplasmic actin". The images from Figure 1 show substantial cytoplasmic actin.

We will show better images to support this.

Measurements are still needed for cytoplasmic actin at various distances from the centrosome

We have measured non-centrosomal cytoplasmic actin in HeLa cells at the same time we see the actin accumulation on centrosomes and found no significant change compared to metaphase levels. We have added this data to further corroborate the specificity of actin accumulation during anaphase.

"As previously noted, CK666 did not have a profound impact on the formation of the mitotic cortical actin mesh." Is there a specific reason to suspect that only profound changes in actin would result in meaningful changes of cellular physiology? Cortical F-Actin should be measured in untreated and CK666 treated cells.

We have removed this statement as it is irrelevant to our analysis, which is focused on noncortical actin.

Referee #4:

I think that the authors have addressed satisfactorily the concerns of reviewer 1 and 2 and this has definitely improved very much the manuscript providing more convincing data and a better quality of analysis.

Concerning reviewer 3, I found that several of the criticisms were not really scientifically sound and I support the authors when they disagree or state they are not correct. However the authors did also reply correctly to other specific comments and this also has improved their manuscript.

Overall I think that the results are supported by the data presented. However what is not entirely clear to me is how relevant this increase in actin nucleation at the centrosome (still relatively mild although apparently specific to anaphase) is. There are no experimental data that really address this issue. I am of course fully aware that this would not be trivial or maybe even doable in a convincing way at the moment. But this leaves the whole story as a quite descriptive work and speculations on possible functional implications. The proposal from the authors is that the increase of actin nucleation at the centrosome in anaphase may be important to lower microtubule nucleation and help in the establishment of the telophase microtubule network. It may be the case but this is really quite a speculation at this point.

In summary I think that the authors answered correctly to the reviewers, improving their manuscript. Still the main concerns of reviewer 1 and 3 is about the relevance of the data, reviewer 1 phrases it as relevance for 'physiological mitosis', reviewer 3 is not convinced by the conclusion that actin nucleation aids the disassembly of centrosomal microtubules. These issues are both pointing at the lack of functional data and were not addressed by the authors.

We thank the reviewer for their comments. We have now included data on Arp2.3 inhibition during regular bipolar mitosis which shows a trend similar to monopolar mitosis. We find that while there is a reduction in actin around the centrosomes during anaphase, the microtubule decrease is also altered (fixed cells). However, we acknowledge the fact that we are concluding based on levels of actin and tubulin, rather than their determining their polymerisation rates. Therefore, we have modified our discussion to reflect this.

Accepted

4th April 2019

Thank you for submitting your final revised manuscript, which I have now been able to consider. I am pleased to inform you that we have now accepted it for publication in The EMBO Journal.

Corresponding Author Name: Buzz Baum

Journal Submitted to: EMBO

Manuscript Number: EMBOJ-2018-99843R